# The epigenetic regulator LSH maintains fork protection and genomic stability via MacroH2A deposition and RAD51 filament formation

Xiaoping Xu[1], Kai Ni[1], Yafeng He[1], Jianke Ren[1], Chongkui Sun[2], Yie Liu[2], Mirit I. Aladjem [3], Sandra Burkett [4], Richard Finney[5], Xia Ding[6,8], Shyam K. Sharan [6] & Kathrin Muegge [1,7✉]

The Immunodeficiency Centromeric Instability Facial Anomalies (ICF) 4 syndrome is caused by mutations in LSH/HELLS, a chromatin remodeler promoting incorporation of histone variant macroH2A. Here, we demonstrate that LSH depletion results in degradation of nascent DNA at stalled replication forks and the generation of genomic instability. The protection of stalled forks is mediated by macroH2A, whose knockdown mimics LSH depletion and whose overexpression rescues nascent DNA degradation. LSH or macroH2A deficiency leads to an impairment of RAD51 loading, a factor that prevents MRE11 and EXO1 mediated nascent DNA degradation. The defect in RAD51 loading is linked to a disbalance of BRCA1 and 53BP1 accumulation at stalled forks. This is associated with perturbed histone modifications, including abnormal H4K20 methylation that is critical for BRCA1 enrichment and 53BP1 exclusion. Altogether, our results illuminate the mechanism underlying a human syndrome and reveal a critical role of LSH mediated chromatin remodeling in genomic stability.

[1] Epigenetics Section, Mouse Cancer Genetics Program, National Cancer Institute, Frederick, MD, USA. [2] Laboratory of Molecular Gerontology, National Institute on Aging, NIH, Baltimore, MD, USA. [3] Developmental Therapeutics Branch, Center for Cancer Research, National Cancer Institute, National Institutes of Health, Bethesda, MD, USA. [4] Molecular Cytogenetics Core Facility, MCGP, CCR, National Cancer Institute, NIH, Frederick, MD, USA. [5] CCR Collaborative Bioinformatics Resource, Center for Cancer Research, National Cancer Institute, Bethesda, MD, USA. [6] Genetics of Cancer Susceptibility Section, Mouse Cancer Genetics Program, National Cancer Institute, Frederick, MD, USA. [7] Basic Science Program, Frederick National Laboratory for Cancer Research, Leidos Biomedical Research, Inc., Frederick, MD 21702, USA. [8] Present address: Oncology R&D, Pfizer, Inc., La Jolla, CA, USA. ✉email: Kathrin.Muegge@nih.gov

Faithful replication of DNA is critical for genome integrity[1–3]. Replication forks can encounter numerous obstacles that impede the replication process. Regions in the human genome that are inherently difficult to replicate, include chromosome fragile sites, telomeres, and other loci with repeat sequences[2]. These loci share some features, such as the formation of secondary DNA structures, tightly bound protein complexes, packaging into heterochromatin, or increased risk of DNA polymerase slippage, and pose challenges to the replisome which can lead to replication stalling[4,5]. Upon stalling, the replication machinery can dissociate, and fork reversal can occur[6]. Reversed forks can be restarted but also undergo resection by nucleases[6]. Unresolved fork stalling, fork collapse, and degradation may lead to the generation of double-stranded DNA breaks (DSBs) and an increase in chromosomal aberrations[5].

A multitude of proteins, including BRCA1, BRCA2, and RAD51, is required for replication fork protection as well as homology-directed repair of DSBs[5–9]. In the context of homology-directed DNA repair BRCA1 is counterbalanced by the repair factor 53BP1, and both factors may occupy sites of DNA damage in a mutually exclusive relationship[1,9]. Under replication stress conditions which lead to fork stalling, BRCA1/2 proteins facilitate RAD51 filament formation which in turn prevents the degradation of newly synthesized DNA strands by nucleases[6,7,10]. The pathological degradation of replication intermediates in the absence of BRCA1/2 proteins or of RAD51 leads to increased genomic instability. Some DNA repair promoting factors bind directly to DNA, while others interact with specific histone modifications[11]. Though it is hypothesized that chromatin compaction or relaxation modulates the accessibility of appropriate repair factors at stalled replication forks[1,11], it remains to be determined what specific features of the chromatin environment are favorable for their assembly and how this prevents the degradation of nascent DNA.

Immunodeficiency, centromeric instability, facial anomalies (ICF) 4 is a severe human disease caused by mutations of the *LSH/HELLS* gene[12]. ICF4 patients suffer from a myriad of symptoms, which are in part phenocopied by *Lsh* knockout mice, including immunodeficiency, neurologic defects, and reduced growth[12–16]. LSH is an ATP-dependent remodeler that contributes to genome-wide incorporation of macroH2A1 and macroH2A2[17]. The two histone variants macroH2A1 and macroH2A2 are encoded by two different genes and their products show a similar distribution pattern in the genome[18–20]. Both variants confer nucleosome stability and promote chromatin compaction[18,19,21]. A few factors have been found to remove macroH2A from its genomic location, such as ATRX which reduces macroH2A occupancy at telomeres, or FACT which removes macroH2A from transcriptionally active regions[20]. LSH is so far the only known factor involved in genome-wide incorporation, but it is currently unknown how LSH interacts with either variant[17,20]. LSH is also known to influence DNA methylation level and to represses transcription of diverse repeat sequences, including ribosomal genes and retrotransposons[15,22,23]. LSH has been implicated in DNA repair processes in response to ionizing radiation[24] and in promoting the efficiency of non-homologous end joining[25].

An important hallmark of the ICF syndrome is genome instability with chromosomal aberrations in activated lymphocytes[12,26]. While signs of DNA damage can accumulate in cell lines derived from ICF4 patients[25], key questions remain: what is the nature of the genomic instability in the ICF4 syndrome and by which molecular mechanism does LSH protect the genome? In this study, we investigated how genomic instability arises in LSH-deficient cells. We found that LSH-deficient cells are more susceptible to replication fork stalling and show increased signs of DNA damage. LSH is critical for the protection of nascent DNA at stalled forks which is mediated by macroH2A deposition. LSH maintains a chromatin environment at the stalled fork that allows for the recruitment of factors involved in the protection of nascent DNA. Loss of either LSH or macroH2A leads to a decrease in RAD51 filament formation which in turn results in increased nucleolytic degradation of nascent DNA tracts by nucleases. Compromised RAD51 loading is associated with an imbalance of BRCA1 and 53BP1 accumulation at the stalled fork and disarray of histone modifications critical for BRCA1 recruitment and exclusion of 53BP1. Collectively, our results suggest a functional requirement for LSH-mediated macroH2A deposition in preserving a chromatin environment pivotal for genomic stability.

## Results

**Proliferation-associated genomic instability in LSH-deficient cells.** Cells obtained from ICF4 patients exhibit signs of genomic instability[12,16], such as chromosomal aberrations, but the precise nature of genomic instability remains largely unknown. We used conditional *Lsh* knockout mice with *Lsh* gene depletion in the hematopoietic system to investigate signs of genomic instability in LSH-KO and control lymphocytes (Supplementary Fig. 1a)[14]. Using immunofluorescence analysis, we found that LSH-deficient lymphocytes, stimulated with either B (Fig. 1a) or T cell mitogens (Supplementary Fig. 1b), displayed elevated levels of γH2AX, a marker of DNA damage which can be also induced by replication fork stalling[27]. The frequency of spontaneous sister chromatid exchange, a sensitive assay to detect recombination which can be associated with replication stress[28] was raised in LSH-depleted lymphocytes compared with LSH-proficient cells (Fig. 1b, Supplementary Fig. 1c). The frequency of chromosomal aberrations, which can form in response to replication stress[3], was significantly elevated in LSH-deficient lymphocytes (Fig. 1c, Supplementary Fig. 1d). Telomere fragility that can be induced by replication stress[29] was increased in LSH-depleted lymphocytes compared with controls (Fig. 1d, Supplementary Fig. 1e). Genomic instability, however, was not associated with substantial increases in cell death, since LSH-deficient B cells display no significant differences in cellular growth and survival rates in vitro when compared to LSH-proficient cells[14]. Altogether, our data revealed increased signs of DNA damage and genomic instability in proliferating LSH-deficient lymphocytes.

To test whether depletion of LSH makes cells more susceptible to replication stress, we depleted LSH in human U2OS cells by RNA interference and exposed them either to hydroxyurea (HU), a drug that induces replication fork stalling through rapid depletion of dNTPs[30], or to aphidicolin (APH), a drug that inhibits DNA polymerase[31]. When we performed clonogenic survival assays in the presence of either HU or APH, we observed impaired colony formation capacity in LSH-deficient cells compared to control cells (Supplementary Fig. 3a–d). This suggested that LSH-deficient cells are more susceptible to replication stress. We also detected an increase of γH2AX and phosphorylation of RPA2 (S4/S8), mediated by DNA-PK, ATR, and ATM in response to DNA damage[32], in LSH-deficient cells compared with controls (Supplementary Fig. 4a, b). Using γH2AX chromatin immunoprecipitation (ChIP), we examined chromosomal fragile sites, which are specific genomic loci that preferentially exhibit gaps and breaks on metaphase chromosomes following partial inhibition of replication[33]. In particular, we looked at common fragile sites (CFSs), which are the largest class of fragile sites, and which can be induced in cell culture by low doses of APH. We detected enhanced enrichment of γH2AX at CFSs after APH treatment in LSH-deficient cells compared with controls (Fig. 1e). In addition, we observed an increase of γH2AX at satellite sequences upon APH treatment (Supplementary Fig. 4c), another region in the genome which is difficult to replicate and where

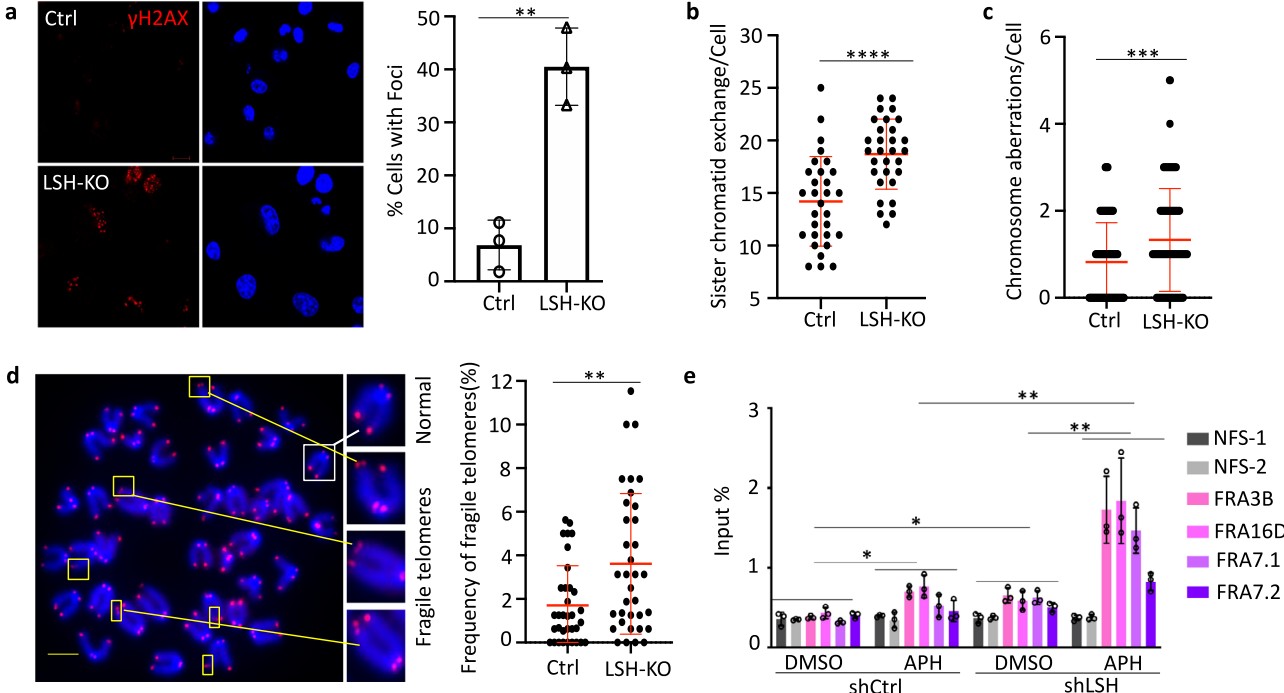

**Fig. 1 Signs of genomic instability and vulnerability to replication stress. a** The image shows γH2AX staining in lipopolysaccharide (LPS) stimulated LSH depleted (LSH-KO) or control (Ctrl) lymphocytes. The bar graph shows the percentage of γH2AX positive cells with more than 5 foci. Data are represented as mean ± SD ($n = 3$ biologically independent experiments). The scale bar represents 10 μm. **$p < 0.01$ by Student's two-tailed $t$ test. **b** Sister-chromatid exchange (SCE) to metaphase chromosome from LPS stimulated LSH depleted (LSH-KO) or control (Ctrl) lymphocytes were determined by Giemsa staining and light microscopy. Representative of $n = 3$ biologically independent experiments. Data are represented as mean ± SD ($n = 30$ cells). Representative images are shown in Fig. S1c. ****$p < 0.0001$ by Student's two-tailed $t$ test. **c** Chromosome aberrations to metaphase chromosome from LPS-stimulated LSH-KO or Ctrl lymphocytes were determined by Giemsa staining. The main chromosomal aberrations included breaks, gaps, arm losses, and dicentric translocation. Representative of $n = 5$ biologically independent experiments. Data are represented as mean ± SD ($n = 140$ cells). The representative images are shown in Fig. S1d. ***$p < 0.001$ by Student's two-tailed $t$ test. **d** Telomeres were identified by fluorescence in situ hybridization (FISH) with telomeric repeat probes. The red dots show the telomere signal in lymphocytes. The white rectangles indicate the normal telomere, while the yellow rectangles indicate fragile telomeres with either multi-telomeric signals or elongated telomeres. A blow-up of the same image is shown in Supplementary Fig. 2. Representative images of chromosomes from control cells are shown in Fig. S1e. Representative of $n = 3$ biologically independent experiments. Data are represented as mean ± SD ($n = 38$ cells). The scale bar represents 5 μm. **$p < 0.01$ by Student's two-tailed $t$ test. **e** ChIP-qPCR analysis for detection of γH2AX enrichment at common fragile sites (FRA3B, FRA16D, FRA7.1, and FRA7.2) and control sites (NFS-1, NFS-2) in mock-treated (DMSO) or aphidicolin-treated (APH) U2OS cells transfected with shCtrl or shLSH. Data are represented as mean ± SD ($n = 3$ independent experiments). *$p < 0.05$, **$p < 0.01$ by Student's two-tailed $t$ test.

replication stress is pronounced[2]. To confirm that a rise in γH2AX is associated with DNA damage, cells were subjected to single-cell electrophoresis where comet tails indicate chromatin relaxation as a result of DNA breaks[34]. LSH-deficient cells showed a significant increase in tail length compared with controls signaling a rise in DNA damage upon HU treatment (Supplementary Fig. 4d). Collectively, LSH-deficient cells are vulnerable to replication stress and display increased signs of genomic instability in response to replication fork inhibitors.

**LSH is critical to protect nascent DNA at stalled replication forks**. To gain insight into the mechanism of how LSH prevents the formation of DNA damage in response to replication stress, we performed DNA fiber analysis, a method that allows one to examine genome-wide replication fork dynamics at single-molecule resolution[35]. Cells were sequentially labeled with two thymidine analogs, 5-chloro-2′-deoxyuridine (CldU, red) and 5-iodo-2′-deoxyuridine (IdU, green) followed by HU treatment which leads to replication fork stalling (Fig. 2a). Stalled forks can reverse and become vulnerable to degradation by nucleases. The ratio of IdU tract length over CldU indicates perturbation of the replication fork and serves as a measure for nucleolytic degradation of nascent DNA[36–39]. Wild type U2OS cells displayed an

IdU/CldU ratio close to 1 indicating that the integrity of stalled forks is not compromised during a short period of replication stress in the presence of LSH (Fig. 2b). However, LSH-deficient U2OS cells exhibited a 45% reduction of the ratio when challenged with HU ($p < 0.0001$; Mann–Whitney test) (Fig. 2b, Supplementary Fig. 5a). The CldU tract length was not significantly changed (Fig. 2c) and the percentage of EdU positive cells was similar comparing LSH-deficient cells with wild type controls (Supplementary Fig. 5b). Furthermore, we found a significantly reduced IdU/CldU ratio in LSH-KO lymphocytes (Fig. 2d) suggesting that susceptibility to spontaneous replication stress may be a source of genomic instability. In addition, murine LSH-deficient embryonic stem (ES) cells showed signs of replication stress. The CldU tract length showed a small increase, possibly indicating differences in embryonic cells, but the ratio of IdU/CldU was also significantly reduced in LSH-deficient ES cells compared to controls suggesting degradation of nascent DNA tracts (Supplementary Fig. 5c–f). Altogether, our data demonstrate that LSH is important for the protection of nascent DNA at stalled forks in diverse cell types.

Extended fork stalling impairs the ability of fork recovery and replication restart[40]. To understand the fate of replication forks in LSH-deficient cells and to assess restart, we pulse-labeled U2OS

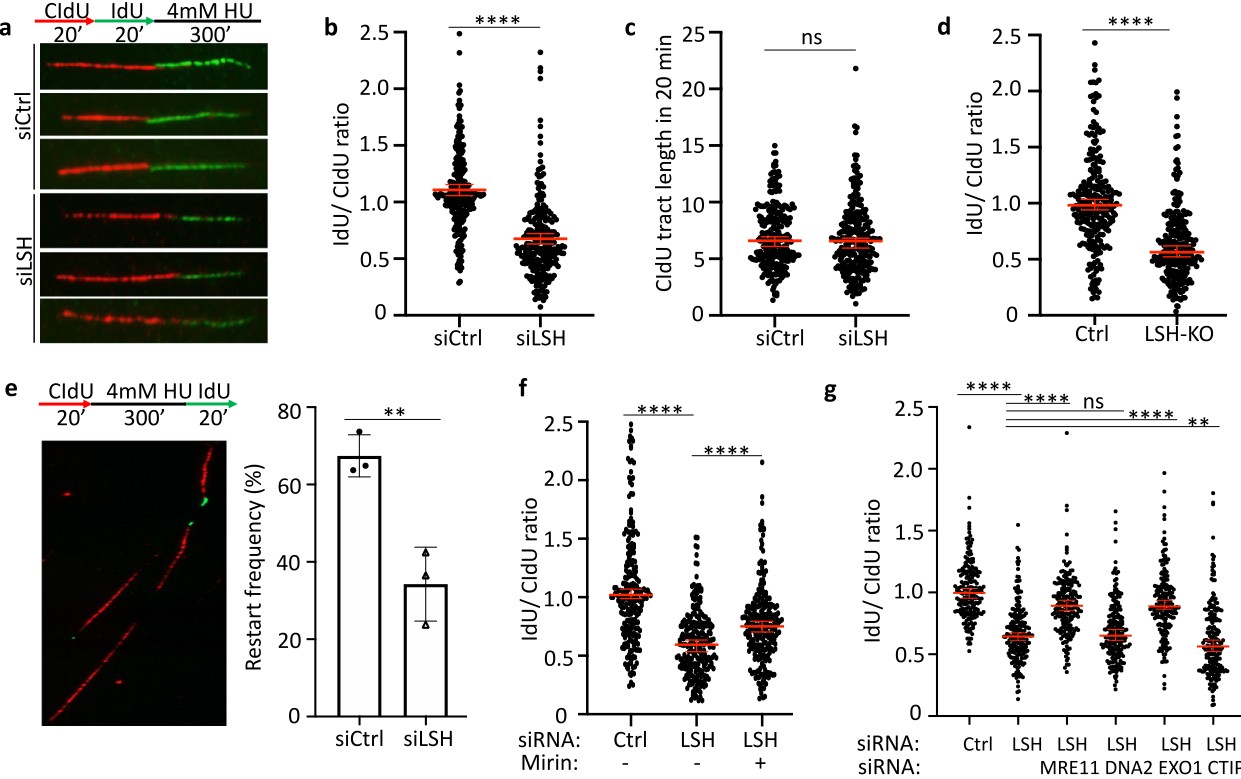

**Fig. 2 Reduced nascent DNA protection at stalled replication forks in LSH-deficient cells. a** Schematic diagram of the fork protection assay (upper panel) and representative images of protected or resected nascent DNA from U2OS cells (lower panel). Cells were sequentially labeled in the indicated times with 5-chloro-2′-deoxyuridine (CldU, red) and 5-iodo-2′-deoxyuridine (IdU, green) followed by 4 mM HU exposure. The ratio of IdU/CldU serves as a measure for nucleolytic degradation of nascent DNA. **b** Nascent DNA degradation analysis in U2OS cells transfected with small interfering RNA (siRNA) to deplete LSH (siLSH) or using control siRNA (siCtrl). Representative of $n = 3$ independent experiments. Data are represented as median ± 95% CI ($n = 250$ DNA fibers). ****$p < 0.0001$ by two-tailed Mann–Whitney test. **c** CldU tract length analysis (20 min CldU incorporation) in U2OS cells transfected with LSH siRNA (siLSH) or control siRNA (siCtrl). Representative of $n = 3$ independent experiments. Data are represented as median ± 95% CI ($n = 250$ DNA fibers). ns = not significant by two-tailed Mann–Whitney test. **d** Nascent DNA degradation analysis in lipopolysaccharide (LPS)-stimulated LSH-depleted lymphocytes (KO) or control lymphocytes (Ctrl). Representative of $n = 3$ biologically independent experiments. Data are represented as median ± 95% CI ($n = 250$ DNA fibers). ****$p < 0.0001$ by two-tailed Mann–Whitney test. **e** Schematic diagram of fork restart assay (left upper panel) and a representative image of stalled (red only) or restarted (red-green) nascent DNA (left lower panel). Fork restart analysis from U2OS cells transfected with the indicated siRNA. Data are represented as mean ± SD ($n = 3$ independent experiments). **$p < 0.01$ by Student's two-tailed $t$ test. **f** Nascent DNA degradation analysis in U2OS cells transfected with the indicated siRNA in the presence or absence of 50 μM Mirin. Representative of $n = 3$ independent experiments. Data are represented as median ± 95% CI ($n = 250$ DNA fibers). ****$p < 0.0001$ by two-tailed Mann–Whitney test. **g** Nascent DNA degradation analysis in U2OS cells transfected with the indicated siRNA to deplete LSH (siLSH) alone or to deplete LSH together with the nuclease MRE11, DNA2, EXO1, or CTIP. Representative of $n = 3$ independent experiments. Data are represented as median ± 95% CI ($n = 200$ DNA fibers). ****$p < 0.0001$ by two-tailed Mann–Whitney test.

cells with CldU for 20 min, blocked them in HU for 5 h, followed by IdU labeling for another 20 min (Fig. 2e). LSH-deficient cells displayed reduced ability to restart compared with control cells (34% versus 67% in controls), suggesting that most forks become inactivated in the absence of LSH upon HU treatment (Fig. 2e). Prolonged stalling can lead to fork reversal and processing by DSBs repair nucleases[40], such as MRE11, a major nuclease present at replication forks in a DNA damage independent manner[41]. To test for a role of MRE11, we used mirin, a chemical inhibitor of MRE11 exonuclease activity[42,43] which had been previously used to protect stalled replication forks[44,45]. With mirin treatment the ratio of IdU/CldU changed from 0.59 to 0.75 in LSH-deficient cells indicating partial recovery of nascent DNA protection (Fig. 2f). This suggested that LSH suppressed access or activity of MRE11, since MRE11 protein levels were unchanged in LSH-deficient cells (Supplementary Fig. 6a). To further corroborate the involvement of MRE11 and to examine the involvement of other major nucleases, such as EXO1, DNA2,

and CTIP[38,40,45], we depleted individual nucleases by RNA interference (Supplementary Fig. 6b, c). While depletion of MRE11 and EXO1 resulted in a partial recovery of nascent DNA in LSH-deficient cells, compared to controls, reduction of DNA2 nuclease had no effect on the restoration of fork degradation (Fig. 2g). These results are consistent with a previous study reporting that MRE11 and EXO1, but not DNA2 mediate nascent strand degradation in BRCA1-deficient cells[45]. Neither did depletion of CTIP resulted in the recovery of nascent DNA at stalled forks (Fig. 2g). Instead, the combination of LSH and CTIP knockdown resulted in a further reduction of the IdU-/CldU-tract length ratio (Fig. 2g). This is consistent with a previous study that reported that CTIP deficiency does not restore fork degradation in BRCA1-mutant cells, but synergistically compromised fork stability suggesting a non-epistatic relation between CTIP and BRCA1[38]. Thus, we concluded, that MRE11 and EXO1 are the major nucleases mediating nucleolytic degradation in the absence of LSH.

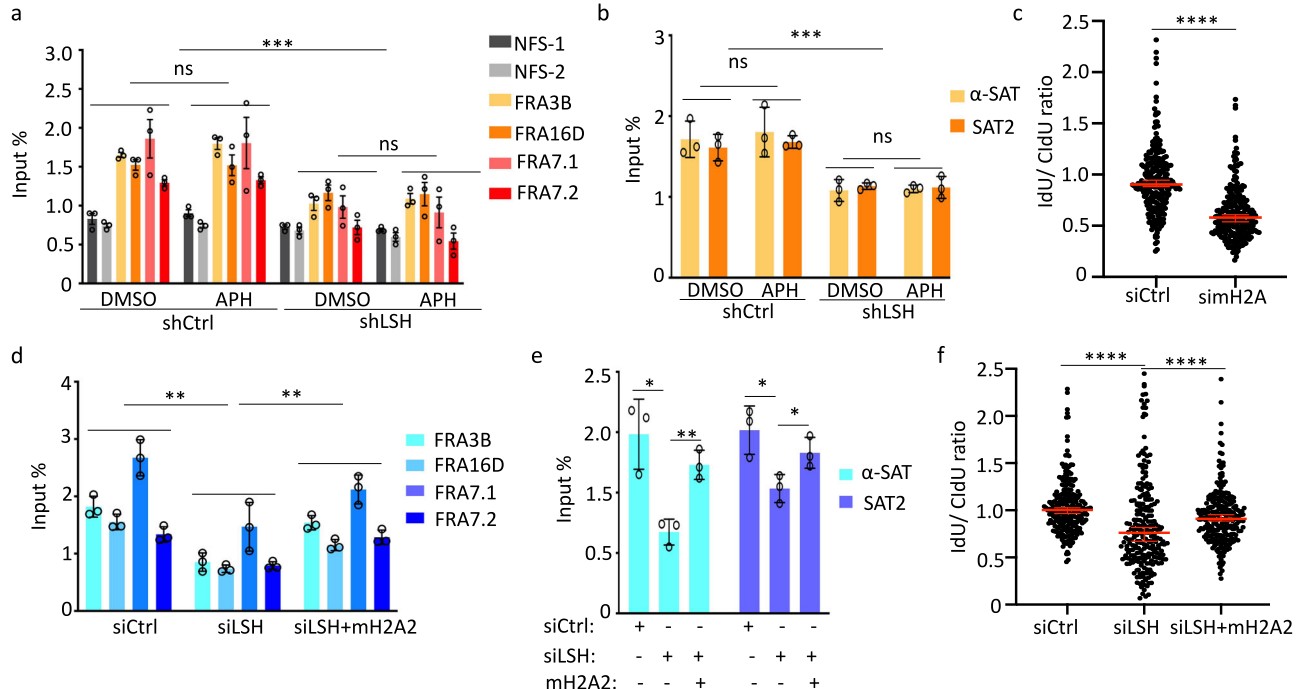

**Fig. 3 MacroH2A deficiency phenocopies LSH deficiency. a** ChIP-qPCR analysis for detection of macroH2A1 enrichment at common fragile sites (FRA3B, FRA16D, FRA7.1, and FRA7.2) and control sites (NFS-1, NFS-2) in mock-treated (DMSO) or aphidicolin-treated (APH) U2OS cells transfected with LSH shRNA (shLSH) or control shRNA (shCtrl). Data are represented as mean ± SD ($n = 3$ independent experiments). ns = not significant, ***$p < 0.001$ by Student's two-tailed $t$ test. **b** ChIP-qPCR analysis for detection of macroH2A1 enrichment at satellite sequences (α-SAT and SAT-2) in mock-treated (DMSO) or aphidicolin-treated (APH) U2OS cells transfected with shLSH or shCtrl. Data are represented as mean ± SD ($n = 3$ independent experiments). ns = not significant, ***$p < 0.001$ by Student's two-tailed $t$ test. **c** Nascent DNA degradation analysis in U2OS cells transfected with the indicated siRNA. The IdU/CldU ratio calculations are done as outlined for Fig. 2a. Representative of $n = 3$ independent experiments. Data are represented as median ± 95% CI ($n = 250$ DNA fibers). ****$p < 0.0001$ by two-tailed Mann–Whitney test. **d** ChIP-qPCR analysis for mH2A2 enrichment at CFSs in untreated U2OS cells transfected with the indicated siRNA (siCtrl or siLSH) and with macroH2A2 expressing construct (mH2A2 + siLSH). Data are represented as mean ± SD ($n = 3$ independent experiments). **$p < 0.01$ by Student's two-tailed $t$ test. **e** ChIP-qPCR analysis for mH2A2 enrichment at satellite sequences (α-SAT and SAT-2) in untreated U2OS cells transfected with the indicated siRNA (siCtrl or siLSH) and with a macroH2A2-expressing construct (siLSH + mH2A2). Data are represented as mean ± SD ($n = 3$ independent experiments). *$p < 0.05$, **$p < 0.01$ by Student's two-tailed $t$ test. **f** Nascent DNA degradation analysis in U2OS cells transfected with the indicated siRNA (siCtrl or siLSH) and with a macroH2A2-expressing construct (siLSH + mH2A2). The IdU/CldU ratio calculations are done as outlined for Fig. 2a. Representative of $n = 3$ independent experiments. Data are represented as median ± 95% CI ($n = 250$ DNA fibers). ****$p < 0.0001$ by two-tailed Mann–Whitney test.

**LSH mediates replication fork protection via MacroH2A deposition.** LSH is a chromatin remodeling protein[46,47] that promotes the incorporation of macroH2A1 and macroH2A2 into chromatin[17]. The two histone variants show substantial overlap in their genome wide distribution[48] and both variants contribute to chromatin compaction[18–20]. In order to understand if macroH2A reduction in LSH-deficient cells could play a role in susceptibility to replication stress, we first examined the relation of LSH-mediated macroH2A deposition and the presence of genomic regions known to be prone to fragility upon stalling using previously published data generated in murine fibroblasts[17]. We noted that syntenic regions of human CFSs[49] and early replicating fragile sites[50] were associated with high macroH2A occupancy (Supplementary Fig. 7a). Examining the fraction of the genome with the largest differences in macroH2A deposition comparing LSH-deficient cells with wild type cells, we found that the genomic regions with the greatest change were associated with a higher frequency of DNA sequences recently shown to be enriched in fragile sites upon fork stalling[50] including DNA transposon, SINE, and satellite sequences (Supplementary Fig. 7b). Correspondingly, the frequency of early replicating fragile sites was increased in the part of the genome showing the biggest macroH2A differences between LSH-deficient cells and controls (Supplementary Fig. 7b). CFSs

did not show increased frequency, suggesting that LSH distinctively affects fragile sites and that it does not preferentially affect CFSs. Nevertheless, the number of macroH2A peaks and macroH2A domains were significantly reduced at early replicating fragile sites as well as CFSs in LSH-deficient cells, making both types of fragile sites potentially more prone to replication stress (Supplementary Fig. 7c, d). Albeit fragile sites differ in specific cell types, these results suggested a possible link between macroH2A reduction and increased susceptibility to replication stress in LSH-deficient cells.

To determine whether LSH effects on the replication fork are mediated by macroH2A deposition in U2OS cells, we first examined macroH2A enrichment at CFSs and repeat sequences, since these sites are prone to replication fork stalling[2] and because we had seen increases in the DNA damage marker γH2AX upon APH treatment in the absence of LSH (Fig. 1e, Supplementary Fig. 4c). Using ChIP analysis in U2OS cells, we found that macroH2A1 and macroH2A2 occupancy was reduced at CFSs (Fig. 3a, Supplementary Fig. 8a) and at satellite repeat sequences (Fig. 3b, Supplementary Fig. 8b) in LSH-deficient cells when compared with control. The total protein amount of either macroH2A1 or macroH2A2 was unaltered in LSH-deficient cells (Supplementary Fig. 8c), suggesting that the reduced occupancy of macroH2A was based on a failure of macroH2A incorporation

into chromatin and not due to a global reduction in macroH2A protein synthesis.

To test for a role of macroH2A in the protection of nascent DNA, we depleted macroH2A1 and macroH2A2 in U2OS cells by small interfering RNA (siRNA) (Supplementary Fig. 8d). We found that macroH2A-deficient cells failed to maintain the integrity of nascent DNA tracts during replication stalling to a similar extent, as we had observed in LSH-deficient cells (Fig. 3c). Thus, macroH2A depletion phenocopied LSH deficiency.

A recent report had shown that overexpression of histone variant H2A.Z can restore its deposition after the loss of SRCAP, a chromatin remodeler mediating incorporation of histone variant H2A.Z[51]. Thus, we hypothesized that overexpression of macroH2A may restore macroH2A occupancy and may lead to recovery of nascent DNA protection in LSH-deficient cells. When we overexpressed macroH2A2 (Supplementary Fig. 8e) we found that overexpression was able to restore macroH2A2 enrichment at CFSs and satellite repeats to near wild type levels (Fig. 3d, e). Importantly, macroH2A overexpression led to a partial rescue of fork degradation in LSH-deficient cells (Fig. 3f). Our data suggest that LSH's protective role of nascent DNA at replication forks is, in part, mediated by macroH2A.

**LSH is required for RAD51 deposition at the stalled replication fork.** RAD51 is a key protective factor of genome stability since RAD51 forms nucleofilaments that stabilize stalled replication forks, supports restart, and protects nascent DNA from MRE11-dependent degradation[8,27,41]. To reveal the molecular mechanisms by which LSH protects forks, we first examined RAD51 chromatin association under HU treatment. Immunoblot analysis revealed that the association of RAD51 with chromatin was reduced in LSH-deficient cells compared with wild type cells, while the total amount of RAD51 protein was unchanged (Supplementary Fig. 9a, b). This suggested that LSH modulates the loading of RAD51 to chromatin but does not disturb RAD51 protein expression. To further evaluate the contribution of RAD51 on LSH mediated fork protection, we directly monitored events at nascent DNA utilizing a proximity ligation assay (PLA)[52]. Nascent DNA was labeled with EdU and chemically conjugated with biotin enabling subsequent detection of RAD51 association with nascent DNA. The PLA signal (PLA/EdU positive cells) was significantly reduced in LSH-deficient cells compared with controls (Fig. 4a). Specificity of the PLA signal was confirmed by the omission of EdU or reduction of RAD51 protein through siRAD51 interference, which both completely abolished the PLA signal (Supplementary Fig. 10a–c). To corroborate our finding of diminished RAD51 loading at the replication fork we used isolation of proteins on nascent DNA (iPOND) analysis to purify proteins associated with replication forks[53]. Control cells showed RAD51 accumulation at stalled replication forks, whereas LSH-deficient cells displayed diminished RAD51 enrichment at nascent DNA (Fig. 4c), which is consistent with our results from immunofluorescence analysis. Thus, LSH functions to promote RAD51 deposition on newly synthesized DNA at stalled forks.

To explore the events upstream of RAD51 loading, we examined RPA foci formation at nascent DNA in LSH-deficient cells. RPA belongs to the first responders of replication stress, but cannot protect ssDNA on its own[40]. Only displacement by RAD51 prevents MRE11-mediated degradation of nascent DNA at stalled forks[40]. The displacement is preceded by phosphorylation of RPA mediated by ATR[3] which promotes loading of the RAD51 recombinase and recovery of replication fork movement after relief of replication stress[54]. We found increased accumulation of phosphorylated RPA2 (Ser33) at stalled forks in LSH-deficient cells

compared to control cells using the PLA assay (Fig. 4b). Western analysis confirmed the occurrence of phosphorylated RPA2 (Ser33) (Supplementary Fig. 11a). In addition, we found phosphorylation of CHK1 (Ser345) (Supplementary Fig. 11b) which has been shown to depend on ATR activity[3]. Since phosphorylation of CHK1 (Ser345) was decreased upon treatment with a specific ATR inhibitor (Supplementary Fig. 11b), we concluded that the ATR pathway was activated. Notably, neither RPA2 protein amounts nor the total level of RPA2 (Ser33) phosphorylation was increased within 5 h of HU treatment (Supplementary Fig. 11a) suggesting that the early increase of phosphorylated RPA2 (Ser33) in LSH-deficient cells (Fig. 4b) occurred specifically at stalled forks. Thus, the LSH-mediated pathway of nascent DNA protection was interrupted downstream of RPA accumulation and upstream of RAD51 loading.

To address the role of macroH2A proteins in RPA displacement and RAD51 filament formation during perturbed replication we depleted macroH2A (macroH2A1 and macroH2A2). As in LSH-deficient cells, we observed reduced RAD51 loading onto chromatin (Fig. 4d) and a decrease of RAD51 accumulation at the stalled fork in macroH2A-deficient cells compared with controls (Fig. 4f). Furthermore, phosphorylated RPA2 (Ser33) was similarly increased at the stalled fork in macroH2A-depleted cells (Fig. 4g), further supporting a LSH model with a role for macroH2A at the stalled fork.

To determine if RAD51 was functionally involved in replication fork instability in the absence of LSH, we hypothesized that overexpression of RAD51 may protect nascent DNA tracts, as it had been previously shown[55]. Notably, we found that overexpression of RAD51 in LSH-deficient cells (Fig. 4e) could partially rescue replication fork instability upon HU treatment (Fig. 4h) demonstrating that LSH mediated its effect on nascent DNA protection, in part, through RAD51.

The assembly of presynaptic filaments of the RAD51 recombinase is a critical step in replication fork protection as well as homologous directed recombination (HDR), however, both processes are functionally separable[10]. To investigate whether LSH affects also the efficiency of HDR-mediated repair, we deployed a reporter system that expressed green fluorescent protein (GFP) upon successful I-SceI-induced DSBs and HDR-mediated repair[56]. Expression of I-SceI endonuclease allowed for the generation of DSBs at a mutated GFP gene that contained the recognition site for I-SceI endonuclease, and subsequent HDR-directed repair with a GFP fragment located further downstream resulted in GFP expression (Supplementary Fig. 12a). Upon transient expression of a I-SceI vector expression (Supplementary Fig. 12b), flow cytometry analysis displayed similar proportions of GFP positive cells in LSH -deficient cells in comparison to control cells (Supplementary Fig. 12c, d). These results suggested that HDR is not compromised in the absence of LSH and that LSH-mediated RAD51 loading was only critical for replication fork protection.

**RS-1 can rescue fork instability and generation of DNA damage.** Since we had found that overexpression of RAD51 can partially restore stalled forks in LSH-deficient cells (Fig. 4h), we reasoned that the RAD51-stimulatory compound 1 (RS-1) may protect against nascent DNA degradation upon HU treatment in LSH-deficient cells. RS-1 enhances RAD51 filament stability in vitro and promotes the formation of RAD51 foci and homologous recombination activity in vivo[57–59]. Remarkably, RS-1 treatment of LSH-deficient U2OS cells protected partially from HU-induced degradation of replication forks (Fig. 5a), while viability was not affected by RS-1 treatment (Supplementary Fig. 13). Furthermore, RAD51 chromatin association was

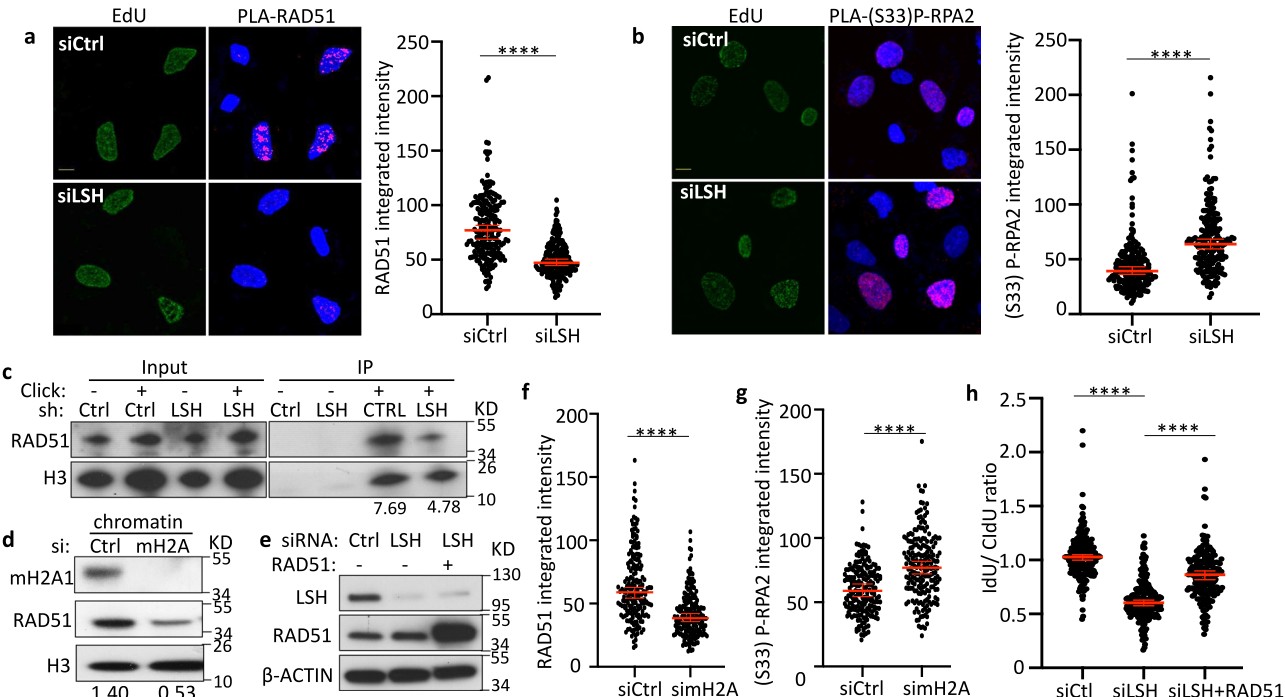

**Fig. 4 Impaired RAD51 deposition at stalled replication forks. a** PLA analysis for U2OS cells transfected with the indicated siRNA (siCtrl or siLSH) and pulsed with 10 μM EdU for 20 min before exposure to 4 mM HU for 5 h. The scale bar represents 10 μm in the representative image. Quantification of PLA signals (RAD51: EdU) is based on EdU-positive cells. Representative of $n = 3$ independent experiments. Data are represented as median ± 95% CI ($n = 200$ cells). ****$p < 0.0001$ by two-tailed Mann–Whitney test. **b** Representative images and quantification of PLA signals (S33P-RPA2: EdU) from U2OS cells transfected with the indicated siRNA after exposure to 4 mM HU for 5 h. The scale bar represents 10 μm. Quantification of PLA signals (S33P-RPA2: EdU) is based on EdU-positive cells. Representative of $n = 3$ independent experiments. Data are represented as median ± 95% CI ($n = 200$ cells). ****$p < 0.0001$ by two-tailed Mann–Whitney test. **c** Immunoblot analysis of iPOND samples to detect EdU-coprecipitated RAD51 protein in U2OS cells transfected with the indicated shRNA (shCtrl or shLSH). 10 μM EdU pulsed for 10 min before the iPOND was performed. Omission of Click-iT chemistry served as negative control and precipitation of H3 as positive control. Normalized Rad51 = (IP-RAD51/IP-H3)/(Input-RAD51/input-H3). **d** U2OS cells were transfected with simH2A (combination of simH2A1 and simH2A2) or control siRNA (siCtrl) and analyzed by immunoblotting for RAD51 protein expression after exposure to 4 mM HU for 5 h using chromatin fractions. **e** U2OS cells were transfected with siCtrl, siLSH, or siLSH in combination with a RAD51 expression construct and analyzed by immunoblotting for RAD51 and LSH protein expression. **f** Quantification of PLA signals (RAD51: EdU) from U2OS cells transfected with the indicated siRNA (siCtrl or simH2A) after exposure to 4 mM HU for 5 h. Quantification of PLA signals (RAD51: EdU) is based on EdU-positive cells. Representative of $n = 3$ independent experiments. Data are represented as median ± 95% CI ($n = 200$ cells). ****$p < 0.0001$ by two-tailed Mann–Whitney test. **g** Quantification of PLA signals (S33P-RPA2: EdU) from U2OS cells transfected with the indicated siRNA (siCtrl or simH2A) after exposure to 4 mM HU for 5 h. Quantification of PLA signals (S33P-RPA2: EdU) is based on EdU-positive cells. Representative of $n = 3$ independent experiments. Data are represented as median ± 95% CI ($n = 200$ cells). ****$p < 0.0001$ by two-tailed Mann–Whitney test. **h** Nascent DNA degradation analysis in U2OS cells transfected with the indicated siRNA (siCtrl or siLSH) and with RAD51 expressing construct (siLSH + RAD51). The IdU/CldU ratio calculations are done as outlined in Fig. 2a. Representative of $n = 3$ independent experiments. Data are represented as median ± 95% CI ($n = 199$ DNA fibers). ****$p < 0.0001$ by two-tailed Mann–Whitney test.

diminished in HU-treated LSH deficient cells compared to controls (Supplementary Fig. 9a, Fig. 5b), but increased upon RS-1 addition (Fig. 5b). Likewise, treatment with RS-1 led to partial recovery of nascent DNA at stalled forks in LSH-depleted lymphocytes (Fig. 5c) and lessened γH2AX increases in LSH-deficient U2OS cells (Fig. 5d). Consistent with these results, RS-1 treatment of LSH-deficient cells suppressed the generation of HU-induced DNA damage to wild type levels (Fig. 5e) and LSH-deficient lymphocytes showed a slight decrease of chromosomal aberrations when cultured in vitro in the presence of RS-1 (Fig. 5f).

Altogether our data indicate improved protection of nascent DNA and increased genomic stability in LSH-deficient cells upon RS-1 treatment which may be due, in part, to a restoration of RAD51 filament formation.

**Imbalance between BRCA1 and 53BP1 loading.** To understand how LSH promotes RAD51 deposition, we examined factors operating upstream of the RAD51 pathway. Multiple factors have been implicated in nascent DNA protection, including BRCA1 that promotes RAD51 filament formation[3,6,7,40]. We found a significant reduction of BRCA1 occupancy at stalled forks in LSH-deficient cells compared with controls using PLA analysis (Fig. 6a). Using ChIP analysis, we observed a decrease of BRCA1 association at CFS upon APH treatment in LSH-deficient cells in comparison with controls (Fig. 6b) further corroborating our PLA results. Notably, BRCA1 protein levels were not reduced in LSH-deficient cells (Supplementary Fig. 14a) indicating that the decrease of BRCA1 at nascent DNA or CFS was site-specific. Thus, impaired BRCA1 chromatin association may contribute to a failure of RAD51 loading upon fork stalling.

BRCA1 counterbalances 53BP1 in pathway choice between homologous recombination versus non-homologous end joining and both factors are in a crosstalk at stalled forks[1,7,9,60]. In a mutually antagonistic relationship, BRCA1 reduces 53BP1 recruitment and 53BP1 appears to minimize the accumulation of BRCA1. Using the PLA assay, we found that 53BP1 occupancy at stalled forks was elevated in LSH-deficient cells compared with

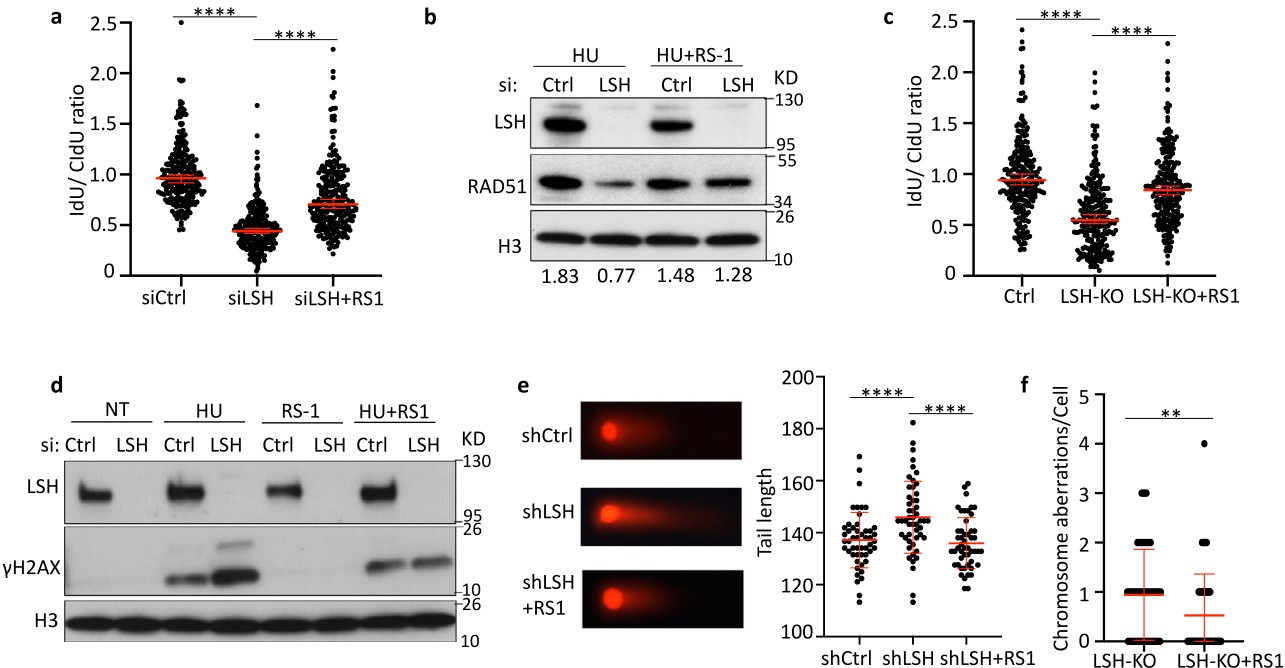

**Fig. 5 Recovery of nascent DNA degradation upon RS-1 treatment. a** Nascent DNA degradation analysis in U2OS cells transfected with the indicated siRNA after exposure to HU in the presence or absence of 10 μM RS-1. The IdU/CldU ratio calculations are done as outlined for Fig. 2a. Representative of $n = 3$ independent experiments. Data are represented as median ± 95% CI ($n = 250$ DNA fibers). ****$p < 0.0001$ by two-tailed Mann–Whitney test. **b** Immunoblot analysis of chromatin fractions for the detection of RAD51 protein. U2OS cells were transfected with the indicated siRNA and exposed to 4 mM HU for 5 h in the presence or absence of 10 μM RS-1. **c** Nascent DNA degradation analysis in LSH-deficient lymphocytes (KO) or control lymphocytes (Ctrl) after exposure to HU in the presence or absence of 10 μM RS-1. The IdU/CldU ratio calculations are done as outlined for Fig. 2a. Representative of $n = 3$ biologically independent experiments. Data are represented as median ± 95% CI ($n = 250$ DNA fibers). ****$p < 0.0001$ by two-tailed Mann–Whitney test. **d** Immunoblot analysis of chromatin fractions for the detection of γH2AX. U2OS cells transfected with the indicated siRNA were analyzed by immunoblotting after exposure to 2 mM HU for 24 h in the presence or absence of 10 μM RS-1. **e** DNA damage was evaluated by neutral comet assay in U2OS cells transfected with the indicated shRNA after exposure to 2 mM HU for 24 h in the presence or absence of 10 μM RS-1. Left: representative images. Right: A scatter plot of tail length from comet assay. Representative of $n = 3$ independent experiments. Data are represented as mean ± SD ($n = 50$ cells). ****$p < 0.0001$ by Student's two-tailed t test. **f** Chromosome aberrations to metaphase chromosome from LPS stimulated LSH-depleted lymphocytes (LSH-KO) in the presence or absence of 10 μM RS-1 was determined by Giemsa staining. Representative of $n = 3$ biologically independent experiments. Data are represented as mean ± SD ($n = 54$ cells). **$p < 0.01$ by Student's two-tailed t test.

controls (Fig. 6c, d). The 53BP1 PLA signal did not coincide with 53BP1 foci by conventional immunofluorescence, since the 53BP1 foci were prevalent in cells that were not in S-phase (Supplementary Fig. 15a). Knockdown of 53BP1 by RNA interference confirmed the specificity of the assay since 53BP1 depletion abrogated the PLA signal (Supplementary Fig. 15b, c). Furthermore, using ChIP analysis, we assessed 53BP1 recruitment at CFSs upon APH treatment and found an increase in LSH-deficient cells compared with controls (Fig. 6e). The increase in 53BP1 binding was not due to changes in protein expression, since 53BP1 protein was expressed at equal amounts in LSH-deficient cells compared to controls (Supplementary Fig. 11a).

Lastly, we considered the possibility that increases in 53BP1 build-up at stalled forks in LSH-deficient cells may be responsible for perturbation of RAD51 filament formation and nascent DNA tract degradation. To test this hypothesis, we lowered 53BP1 protein levels by siRNA interference in LSH-deficient cells to reduce the chance of abnormal 53BP1 accumulation at the stalled fork. We choose an intermediate 53BP1 reduction (Supplementary Fig. 14b), since the complete loss of 53BP1 aggravates nascent DNA degradation[44,61]. We observed significant recovery of replication fork degradation in LSH-deficient cells at reduced 53BP1 protein levels in comparison to LSH-deficient cells with wild type 53BP1 levels (Fig. 6f), suggesting a detrimental role of excess 53BP1 in LSH-mediated fork protection. We also observed a recovery of BRCA1 enrichment at CFSs in LSH-deficient cells

upon 53BP1 depletion (Fig. 6g) which is consistent with the restoration of nascent DNA tract restoration.

Altogether, our data suggest that the exchange of RPA to RAD51 loading was compromised in LSH-deficient cells due to an imbalance of critical 'choice' factors such that BRCA1 was lessened while 53BP1 was incongruously augmented.

**Perturbed chromatin environment at the replication fork.** To understand how the altered chromatin environment induced by LSH loss can lead to a failure to protect nascent DNA, we examined histone variants and histone modifications at the stalled replication fork. Old histones with their modifications are 'recycled' at the replication fork and mixed with new histones in equal ratios on newly replicated DNA[62] leading to a dilution of old histones at a 1:2 ratio. We observed reduced macroH2A1 (Fig. 7a) and macroH2A2 (Fig. 7b) enrichment at stalled replication forks in LSH-deficient cells compared to controls. This reflects in part the overall global reduction in macroH2A occupancy[17], including the decrease at repeat sequences and CFS prior to stalling (Fig. 3a, b, Supplementary Fig. 8a, b). As a control, we examined canonical histone 3 (H3) and found it unaltered at stalled forks in LSH-deficient cells compared to wild-type cells (Fig. 7c). This indicated that the decrease of macroH2A1 and macroH2A2 was histone variant specific. On the other hand, H3K4me3 which accumulates at stalled replication forks in BRCA1-deficient cells[39] was elevated at stalled forks in LSH-deficient cells compared with controls

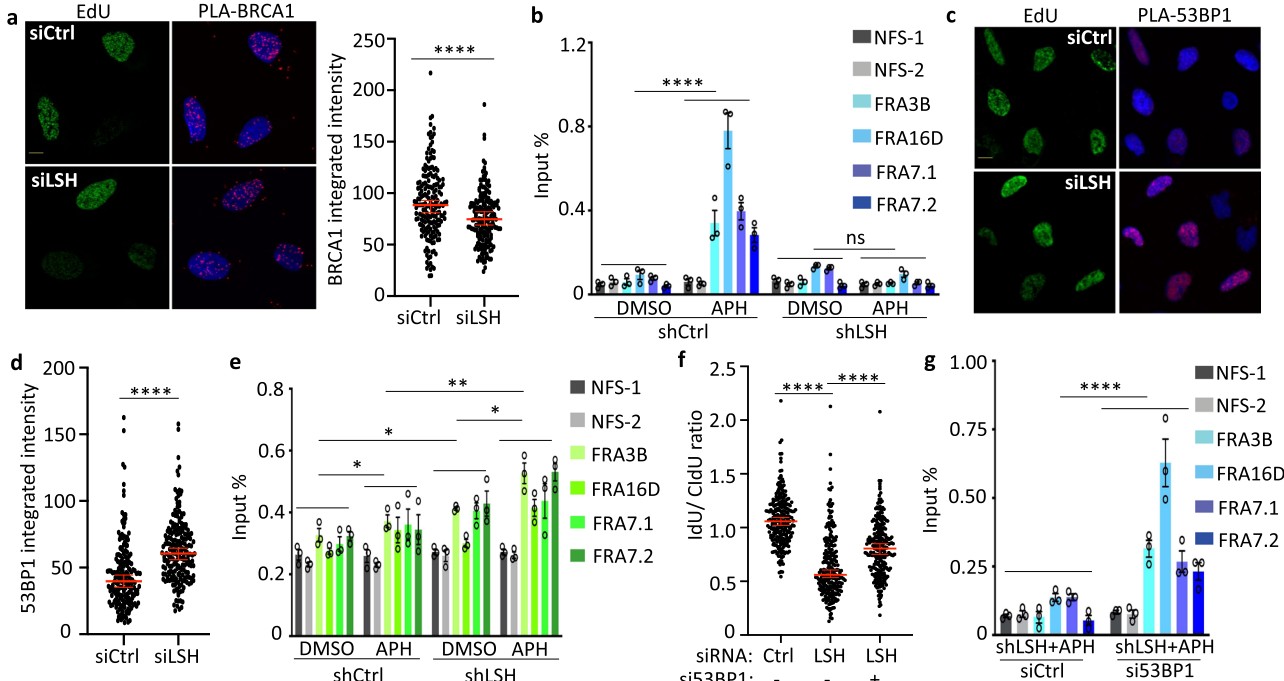

**Fig. 6 Reduced BRCA1 and increased 53BP1 occupancy at stalled replication forks. a** Representative images (left panel) and quantification of PLA signals (BRCA1: EdU) from U2OS cells transfected with the indicated siRNA after exposure to 4 mM HU for 5 h. The scale bar represents 10 μm. Quantification of PLA signals (BRCA1: EdU) is based on EdU-positive cells. Representative of $n = 3$ independent experiments. Data are represented as median ± 95% CI ($n = 200$ cells). ****$p < 0.0001$ by two-tailed Mann–Whitney test. **b** ChIP-qPCR analysis for detection of BRCA1 enrichment at CFSs (FRA3B, FRA16D, FRA7.1, and FRA7.2) and control sites (NFS-1, NFS-2) in mock-treated (DMSO) or aphidicolin-treated (APH) U2OS cells transfected with shLSH or shCtrl. Data are represented as mean ± SD ($n = 3$ independent experiments). ns = not significant, ****$p < 0.0001$ by Student's two-tailed $t$ test. **c** Representative images of PLA signals (53BP1: EdU) from U2OS cells transfected with the indicated siRNA after exposure to 4 mM HU for 5 h. The scale bar represents 10 μm. **d** Quantification of PLA signals (53BP1: EdU) from U2OS cells as treated in Fig. 6C. Quantification of PLA signals (53BP1: EdU) is based on EdU-positive cells. Representative of $n = 3$ independent experiments. Data are represented as median ± 95% CI ($n = 200$ cells). ****$p < 0.0001$ by two-tailed Mann–Whitney test. **e** ChIP-qPCR analysis for detection of 53BP1 enrichment at CFSs (FRA3B, FRA16D, FRA7.1, and FRA7.2) and control sites (NFS-1, NFS-2) in mock-treated (DMSO) or aphidicolin-treated (APH) U2OS cells transfected with shLSH or shCtrl. Data are represented as mean ± SD ($n = 3$ independent experiments). *$p < 0.05$, **$p < 0.01$ by Student's two-tailed $t$ test. **f** Nascent DNA degradation analysis after exposure to HU in U2OS cells transfected with siCtrl or siLSH or a combination of siLSH and si53BP1 leading to an intermediate 53BP1 depletion (0.3 in Supplementary Fig. 13B). The IdU/CldU ratio calculations are done as outlined in Fig. 2a. Representative of $n = 3$ independent experiments. Data are represented as median ± 95% CI ($n = 250$ DNA fibers). ****$p$ value $< 0.0001$ by two-tailed Mann–Whitney test. **g** ChIP-qPCR analysis for detection of BRCA1 enrichment at indicated CFSs in aphidicolin treated (APH) U2OS cells transfected with shLSH and either siCtrl or si53BP1. Data are represented as mean ± SD ($n = 3$ independent experiments). ****$p < 0.01$ by Student's two-tailed $t$ test.

(Fig. 7d). ChIP analysis showed a small increase of H3K4me3 at CFS (Fig. 7e) and satellite repeats (Supplementary Fig. 16a) upon replication stalling in LSH-deficient cells compared with controls. Global H3K4me3 levels were unaltered in LSH-deficient cells (Supplementary Fig. 11a) indicating that H3K4me3 enrichment was replication fork specific. In addition, macroH2A depletion showed a similar increase of H3K4me3 at the stalled replication fork, further supporting the notion that LSH effects are mediated through macroH2A (Fig. 7f). An increase of H3K4me3 at the stalled fork is thought to promote nucleolytic degradation by MRE11[39], which is consistent with a model in which LSH deficiency results in MRE11-dependent degradation of nascent DNA tracts in the absence of LSH.

To evaluate histone modifications critical for BRCA1 and 53BP1 recruitment, we examined dimethylation of H4 at lysine residue K20 (H4K20me2). H4K20me2 is the predominant histone modification in mammalian cells which becomes diluted during replication[63,64]. 53BP1 recognizes H2AK13ub and H4K20me2 containing nucleosomes by its tandem Tudor domains[65] Interestingly, post-replicative chromatin is marked by an unmethylated lysine 20 of H4 (H4K20me0) and the methylation mark H4K20me2 is slowly re-established until G2/M[66]. The recognition

of H4K20me0 by BARD1 is required for recruitment of the BRCA1–BARD1 complex at nascent DNA which leads to BRCA1 accumulation at post-replicative chromatin and exclusion of 53BP1[60,64,67]. Strikingly, we detected an increase in H4K20me2 modification at the replication fork in HU-treated LSH-deficient cells compared to controls (Fig. 8a), while EdU signal intensity was similar comparing LSH deficient to LSH-proficient cells (Fig. 8b). The total amount of this histone modification was unaltered (Supplementary Fig. 11a) suggesting that the increase in H4K20me2 was replication fork specific. MacroH2A depletion resulted also in an increase of H4K20me2 at the stalled replication fork thus mimicking LSH deficiency and suggesting that LSH mediated its effect, in part, through macroH2A (Fig. 8c). ChIP analysis revealed a rise of H4K20me2 at CFS (Fig. 8d) and satellite repeats (Supplementary Fig. 16b) upon replication fork stalling in LSH-deficient cells compared with controls.

Lastly, we reasoned that reducing the level of H4K20me2 may alter the outcome of 53BP1 binding in LSH-deficient cells. The histone methyltransferase KMT5A (also known as PRSET7, SET07, SET8, SETD8) is the only known H4K20 mono methyltransferase and mono-methylation is a prerequisite for subsequent H4K20 di-methylation of post-replicative chromatin[66–68]. Thus, we depleted

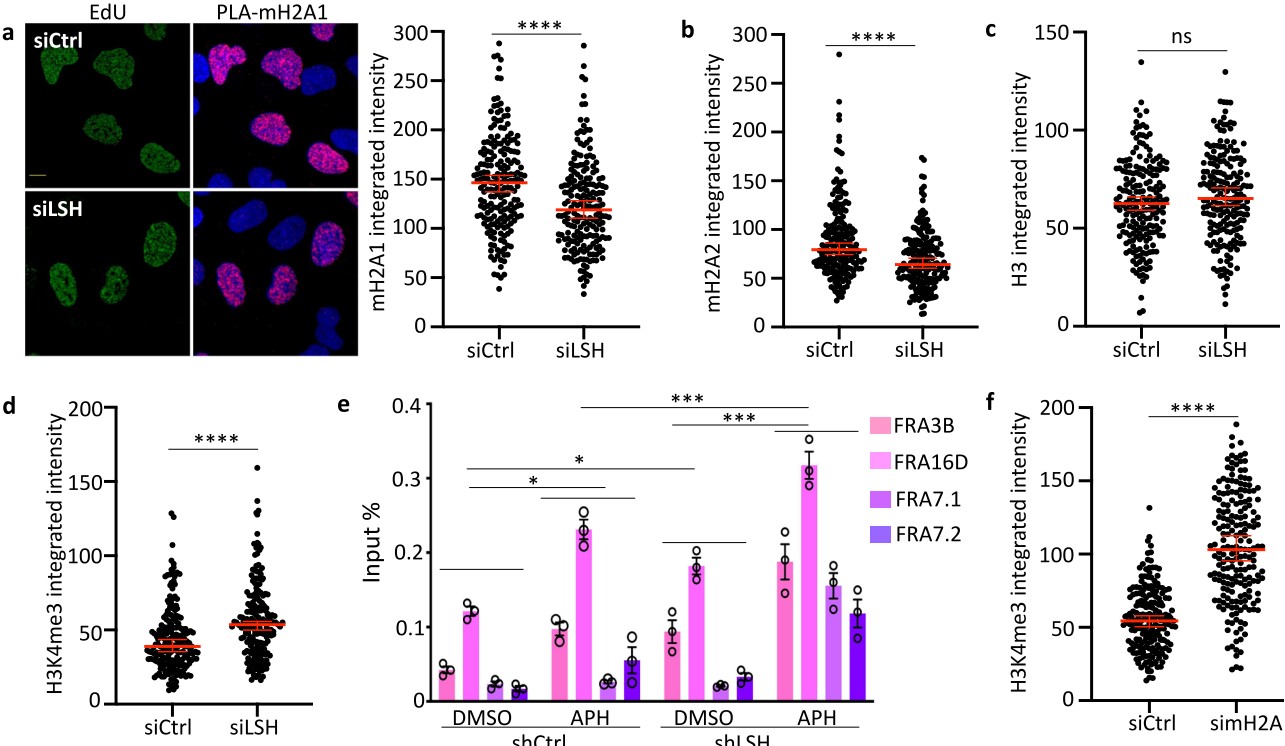

**Fig. 7 Abnormal chromatin environment at stalled forks in LSH-deficient cells. a** Representative images (left panel) and quantification of PLA signals (mH2A1: EdU, right panel) from U2OS cells transfected with the indicated siRNA after exposure to 4 mM HU for 5 h. The scale bar represents 10 μm. Quantification of PLA signals (mH2A1: EdU) is based on EdU-positive cells. Representative of $n = 3$ independent experiments. Data are represented as median ± 95% CI ($n = 200$ cells). ****$p < 0.0001$ by two-tailed Mann–Whitney test. **b** Quantification of PLA signals (mH2A2: EdU) from U2OS cells transfected with the indicated siRNA after exposure to 4 mM HU for 5 h. Quantification of PLA signals (mH2A2: EdU) is based on EdU-positive cells. Representative of $n = 3$ independent experiments. Data are represented as median ± 95% CI ($n = 200$ cells). ****$p < 0.0001$ by two-tailed Mann–Whitney test. **c** Quantification of PLA signals (H3: EdU) from U2OS cells transfected with the indicated siRNA (siCtrl or siLSH) after exposure to 4 mM HU for 5 h. Quantification of PLA signals (H3: EdU) is based on EdU-positive cells. Representative of $n = 3$ independent experiments. Data are represented as median ± 95% CI ($n = 200$ cells). ns = not significant by two-tailed Mann–Whitney test. **d** Quantification of PLA signals (H3K4me3: EdU) from U2OS cells transfected with the indicated siRNA (siCtrl or siLSH) after exposure to 4 mM HU for 5 h. Quantification of PLA signals (H3K4me3: EdU) is based on EdU-positive cells. Representative of $n = 3$ independent experiments. Data are represented as median ± 95% CI ($n = 200$ cells). ****$p < 0.0001$ by two-tailed Mann–Whitney test. **e** ChIP-qPCR analysis for detection of H3K4me3 enrichment at CFSs in mock-treated (DMSO) or aphidicolin-treated (APH) U2OS cells transfected with shLSH or shCtrl. Data are represented as mean ± SD ($n = 3$ independent experiments). *$p < 0.05$, ***$p < 0.001$ by Student's two-tailed $t$ test. **f** Quantification of PLA signals (H3K4me3: EdU) from U2OS cells transfected with the indicated siRNA (siCtrl or simH2A) after exposure to 4 mM HU for 5 h. Quantification of PLA signals (H3K4me3: EdU) is based on EdU-positive cells. Representative of $n = 3$ independent experiments. Data are represented as median ± 95% CI ($n = 200$ cells). ****$p < 0.0001$ by two-tailed Mann–Whitney test.

KMT5A in LSH-deficient cells (Supplementary Fig. 16c) and observed, as expected, lower H4K20me2 level at the stalled replication fork (Fig. 8e). Intriguingly, 53BP1 enrichment at the stalled fork was significantly lowered (Fig. 8f), while BRCA1 enrichment recovered close to control levels (Fig. 8g), further supporting the notion of a balance of the two repair factors which depends on histone modifications.

Altogether, LSH depletion resulted in reduced macroH2A presence at the stalled replication fork and a chromatin environment with histone modifications that are favorable to 53BP1 instead of BRCA1 recruitment and permissive to nucleolytic degradation by nucleases.

## Discussion

The chromatin remodeler LSH plays a critical role in the ICF4 syndrome, but the molecular pathways leading to genomic instability are uncharacterized. Here, the discovery that LSH plays a role in preventing the degradation of nascent DNA strands by promoting RAD51 filament formation helps to identify novel potential targets for treatment of the syndrome. The finding that LSH alters macroH2A deposition and histone modifications at

the stalled fork provides a more complete understanding of how the chromatin environment influences the balance between BRCA1 and 53BP1 binding and modulates the proper pathway choice to maintain genomic integrity in the context of replication fork stalling (model Supplementary Fig. 17).

MacroH2A may be diluted during replication as has been reported for H3 and H4 histones which are approximately equally distributed onto leading and lagging strands[62]. Re-establishment of macroH2A domains is achieved by de novo deposition during the G1 phase of the cell cycle[69]. This is consistent with the hypothesis that LSH exercises its function independent of replication and that the chromatin environment is altered before fork stalling occurs. There is no evidence in our experimental setting that macroH2A occupancy is enriched at the replication fork upon prolonged stalling (Fig. 3a, b, Supplementary Fig. 8a, b). Instead, LSH-deficient cells display a global decrease of macroH2A incorporation into chromatin[17], and therefore macroH2A1 and macroH2A2 show reduced occupancy at the stalled replication fork. Since macroH2A depletion phenocopies LSH deficiency, our results imply that reduced macroH2A occupancy creates a chromatin environment at the stalled fork that promotes

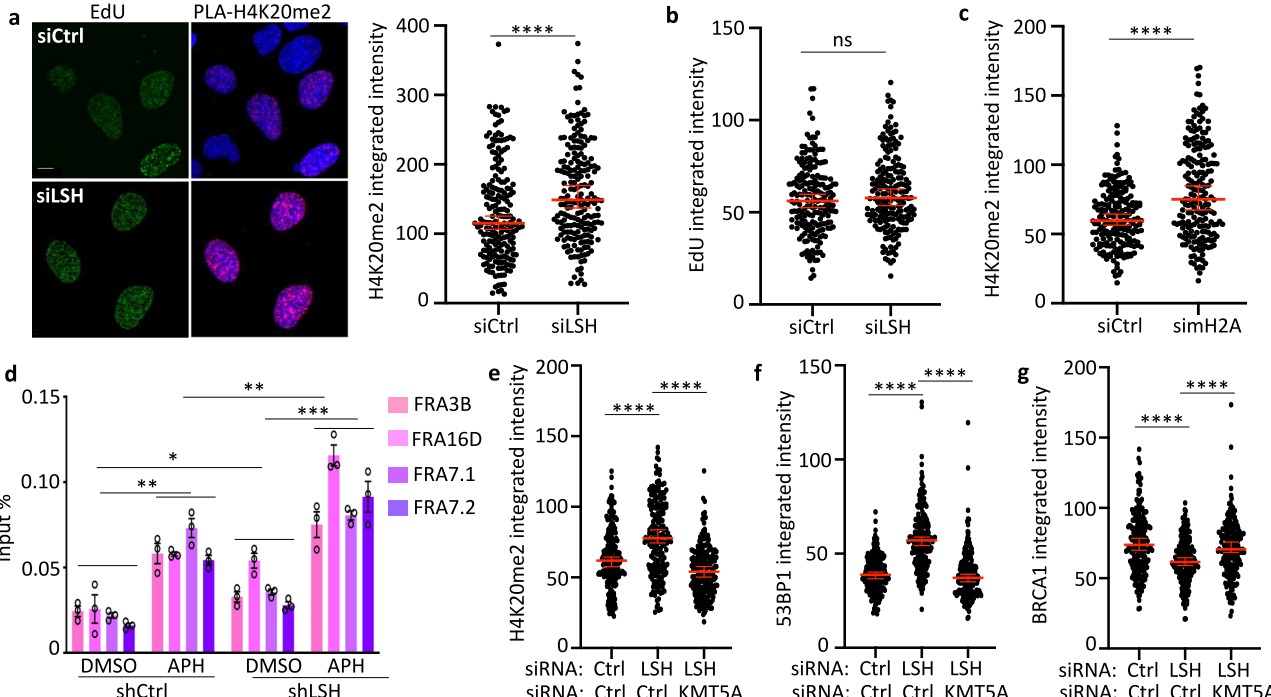

**Fig. 8 A role of H4K20 methylation at stalled forks in LSH-deficient cells. a** Representative images (left panel) and quantification of PLA signals (H4K20me2: EdU) from U2OS cells transfected with the indicated siRNA (siCtrl or siLSH) after exposure to 4 mM HU for 5 h. The scale bar represents 10 μm. Quantification of PLA signals (H4K20me2: EdU) is based on EdU-positive cells. Representative of $n = 3$ independent experiments. Data are represented as median ± 95% CI ($n = 200$ cells). ****$p < 0.0001$ by two-tailed Mann–Whitney test. **b** Quantification of EdU signals from U2OS cells transfected with the indicated siRNA after exposure to 4 mM HU for 5 h. Representative of $n = 3$ independent experiments. Data are represented as median ± 95% CI ($n = 200$ cells). ns = not significant by two-tailed Mann–Whitney test. **c** Quantification of PLA signals (H4K20me2: EdU) from U2OS cells transfected with the indicated siRNA (siCtrl or simH2A) after exposure to 4 mM HU for 5 h. Quantification of PLA signals (H4K20me2: EdU) is based on EdU-positive cells. Representative of $n = 3$ independent experiments. Data are represented as median ± 95% CI ($n = 200$ cells). ****$p < 0.0001$ by two-tailed Mann–Whitney test. **d** ChIP-qPCR analysis for detection of H4K20me2 enrichment at CFSs in mock-treated (DMSO) or aphidicolin-treated (APH) U2OS cells transfected with shLSH or shCtrl. Data are represented as mean ± SD ($n = 3$ independent experiments). *$p < 0.05$, **$p < 0.01$, ***$p < 0.001$ by Student's two-tailed $t$ test. **e** Quantification of PLA signals (H4K20me2: EdU) from U2OS cells transfected with the indicated siRNA (siCtrl or siLSH or siLSH together with siKMT5) after exposure to 4 mM HU for 5 h. Quantification of PLA signals (H4K20me2: EdU) is based on EdU-positive cells. Representative of $n = 3$ independent experiments. Data are represented as median ± 95% CI ($n = 200$ cells). ****$p < 0.0001$ by two-tailed Mann–Whitney test. **f** Quantification of PLA signals (53BP1: EdU) from U2OS cells transfected with the indicated siRNA (siCtrl or siLSH or siLSH together with siKMT5) after exposure to 4 mM HU for 5 h. Quantification of PLA signals (53BP1: EdU) is based on EdU-positive cells. Representative of $n = 3$ independent experiments. Data are represented as median ± 95% CI ($n = 200$ cells). ****$p < 0.0001$ by two-tailed Mann–Whitney test. **g** Quantification of PLA signals (BRCA1: EdU) from U2OS cells transfected with the indicated siRNA (siCtrl or siLSH or siLSH together with siKMT5) after exposure to 4 mM HU for 5 h. Quantification of PLA signals (BRCA1: EdU) is based on EdU-positive cells. Representative of $n = 3$ independent experiments. Data are represented as median ± 95% CI ($n = 200$ cells). ****$p < 0.0001$ by two-tailed Mann–Whitney test.

degradation of nascent DNA. Meanwhile macroH2A is associated with compact chromatin, and it is a possibility that reduced macroH2A at the fork leads to greater accessibility. A recent report studying *S. pombe* cells found that chromatin compaction at the stalled fork protects, whereas chromatin relaxation is associated with greater fork instability[70]. Our model suggests that LSH acts globally on macroH2A deposition, but that an unfavorable pre-lesion chromatin environment at the replication fork promotes fork degradation.

MacroH2A is highly conserved and widespread in vertebrates, and human macroH2A1 and macroH2A2 proteins share 68% identity[20]. Both macroH2A proteins show a similar genome-wide distribution pattern but have distinct biochemical characteristics[18–20,48]. The macroH2A1.1 isoform can interact with poly-ADP-ribosylated proteins and this mechanism is thought responsible for the recruitment of macroH2A1.1 to sites of DSBs[71,72]. However, several findings suggest that this mechanism does not likely contribute to the protection of nascent DNA by LSH. First, the reported recruitment is 'external' and does not lead to the incorporation of macroH2A1.1 into nucleosomes, whereas LSH

promotes histone exchange and macroH2A incorporation into nucleosomes[17]; second, LSH affects the genomic deposition of macroH2A1 as well as macroH2A2, but only macroH2A1.1 can interact with poly-ADP-ribosylated proteins and is recruited to sites of DNA damage[71,72]. Third, we did not observe any increase in macroH2A1 chromatin association upon stalling (Fig. 3a, b and Supplementary Fig. 8a, b); and finally, macroH2A2 overexpression can partially compensate for LSH deficiency (Fig. 3d, e, f and Supplementary Fig. 8e) suggesting that the molecular mechanisms of nascent DNA protection by LSH are independent of 'external' recruitment of macroH2A1.1 and that both histone variants share the ability to protect nascent DNA.

RAD51 filament formation can protect nascent DNA[8,27,41] and we observed reduced RAD51 loading in either LSH-deficient cells or in macroH2A-deficient cells. RS-1, a drug that promotes RAD51 filament formation[57–59], partially restored protection of stalled forks and reduced genomic instability. Moreover, overexpression of RAD51 restored fork protection suggesting that RAD51 plays a key role in LSH-mediated protection of nascent DNA and that RAD51 may serve as a novel target in the therapy

of ICF4 patients. RAD51 loading in turn is promoted by BRCA1, and reduced BRCA1 level at the fork (outlined below) may result in compromised RAD51 loading. In addition, we observed an increase of H3K4me3 at the stalled replication fork in LSH-deficient cells. H3K4me3, which is catalyzed by MLL3/4 histone methyltransferase, promotes MRE11-mediated degradation at stalled forks[39], since depletion of catalytically active MLL3/4 histone methyltransferase reduces the degradation of nascent DNA in BRCA1-deficient cells in a pathway that is currently unknown[39]. This suggests the possibility that in addition to BRCA1 and RAD51 reduction, the increase of H3K4me3 may contribute to fork degradation in LSH-deficient cells.

Our data shows that macroH2A occupancy is decreased at the stalled fork in LSH-deficient cells. MacroH2A is a histone variant that is associated with constitutive heterochromatin and which is depleted from actively transcribed regions[20]. MacroH2A renders nucleosomes more stable and less accessible to exonuclease digestion[18,19,21]. Since macroH2A is associated with compact chromatin, a reduction of macroH2A may promote accessibility for the histone methyltransferase KMT5A (PRSET7, SET07, SET8, SETD8) that mediates methylation of post-replicative H4K20me0[66,68]. It was reported that recognition of H4K20me0 by the BRCA1–BARD1 complex results in BRCA1 enrichment under exclusion of 53BP1[66], while overexpression of KMT5A leads to augmented H4K20me2 and 53BP1 enrichment in S-phase chromatin and reduced BRCA1 recruitment at damaged chromatin[64]. Thus, vanishing H4K20me0 and increases in H4K20me2 in LSH-deficient cells may induce an imbalance of 53BP1 recruitment at the cost of the BRCA1–BARD1 complex that is critical for RAD51 filament formation. This model is consistent with our observation that reduction of the histone methyltransferase KMT5A reduced H4K20me2, increased 53BP1 and reduced BRCA1 enrichment at the stalled replication fork and that lowering of 53BP1 protein level in LSH-deficient cells leads to a partial recovery of replication fork stability. Our model does not exclude the involvement of other pathways. For example, elevated accessibility at the fork in LSH-deficient cells may promote H3K4me3 by MLL3/4 which makes forks more vulnerable to nucleolytic degradation[39]. In addition, other complexes, such as TONSL-MMS22L that require H4K20me0 for recognition and that aid in RAD51 loading to chromatin[66] could be involved in LSH-mediated fork protection. It is noteworthy to mention that normally the degree of H4K20 methylation does not change upon stalling, and that it is thought that chromatin relaxation may additionally unmask the modified histone tails crucial for subsequent interaction and recruitment of repair factors[66]. Therefore, a combination of chromatin de-compression and aberrant histone marks at the replication fork may contribute to compromised RAD51 filament formation and insufficient fork protection in the absence of LSH.

We report here chromatin changes at satellite sequences that are located as repetitive units in pericentromeric and centromeric regions where ICF4 patients show DNA hypomethylation[12]. Centromeric DNA has been shown to generate unusual secondary structures that forms obstacles for the replication machinery and may lead to the slowing of fork progression and spontaneous replication stress[73]. These centromeric secondary structures are processed by specific nuclease activities to allow for normal fork progression[74]. It is conceivable that the defective chromatin organization that we have uncovered here and the hypomethylated state of DNA in the absence of normal LSH/HELLS function may perturb the resolution of these secondary structures and induce increased vulnerability to nucleolytic degradation of nascent DNA in centromeric DNA, thus contributing to genetic instability in the ICF4 syndrome.

In conclusion, we propose a mechanistic model underlying LSH-mediated maintenance of genome integrity. In this model

(Supplementary Fig. 17), a decrease in LSH activity leads to impaired genomic macroH2A deposition and a compromised chromatin environment at the replication fork. The altered composition of histone variant macroH2A and histone modifications, especially the methylation state of H4K20 at stalled replication forks results in aberrant accumulation of 53BP1 and debilitated recruitment of BRCA1. The imbalance of protective factors reduces RAD51 filament formation required for nascent DNA protection against MRE11-mediated and EXO1-mediated nucleolytic cleavage. Thus, the remodeling activity of LSH, with the help of the histone variant macroH2A, modulates sensitivity to replication stress and protects against genomic instability, a mechanism that may contribute to the pathophysiology of the human ICF4 syndrome.

## Methods

**Cell culture and generation of cell lines**. Splenocytes were derived from Cre-recombinase-mediated conditional deletion of Lox-P site flanked ("floxed") *Lsh* gene[14]. Splenocytes were cultured in Roswell Park Memorial Institute (RPMI) 1640 Medium (Thermo Fisher Scientific) supplemented with 10% fetal bovine serum (FBS; Omega Scientific), 25 μg/ml lipopolysaccharide (LPS) (Sigma Aldrich) and standard antibiotics. U2OS cells (ATCC) were cultured in Dulbecco's modified Eagle medium (DMEM; Thermo Fisher Scientific) supplemented with 10% FBS and standard antibiotics. Murine embryonic stem cells (mESCs) were cultured on mitotically inactive SNL feeder cells in M15 media, which is Knockout DMEM media (Thermo Fisher Scientific) supplemented with 15% FBS (Thermo Fisher Scientific), 0.00072% β-mercaptoethanol (Sigma Aldrich), 100 U/ml penicillin (Thermo Fisher Scientific), 100 mg/ml streptomycin (Thermo Fisher Scientific), and 0.292 mg/ml L-glutamine (Thermo Fisher Scientific). U2OS cells or mESCs stably expressing shRNA were generated by lentivirus transfection followed by clonal selection of cells grown in media containing 5 μg/ml puromycin. U2OS cells stably expressing GFP-macroH2A2 fusion proteins were generated by plasmid transfection followed by cell sorting and clonal selection of cells grown in media containing 500 μg/ml neomycin (Thermo Fisher Scientific). MCF10A DR-GFP cells were a kind gift of Prof. M. Jasin and were cultured in DMEM/nutrient mixture F-12 (DMEM/F-12; Thermo Fisher Scientific) supplemented with 5% horse serum (Thermo Fisher Scientific), 20 ng/ml epidermal growth factor, 0.5 mg/ml hydrocortisone, 100 ng/ml cholera toxin, 10 μg/ml insulin, and 1% penicillin–streptomycin.

**Drugs and inhibitors**. HU, APH, Mirin and RS-1 were purchased from Sigma Aldrich. dNTP analogs BrdU, EdU, CldU were from Sigma Aldrich and IdU from MP Biologicals. ATR inhibitor of Berzosertib (VE-822, #S7102) was ordered from Selleck Chemicals.

**Cell viability and colony formation assay**. For the cell viability assay, U2OS stably expressing shRNA were seeded in five repeats at a density of 5000 cells/well in 96-well plate. Cells were exposed to the indicated doses of HU and/or RS-1 at 24 h after plating and grown for 5 h at 37 °C. To measure cell viability, Cell Counting Kit-8 reagent (Dojindo) was added, cells were incubated at 37 °C for 3 h, and fluorescence was measured at 450 nm using a CLARIOstar Microplate Reader (BMG Labtech).

To assess clonogenic survival in response to replication stress, U2OS stably expressing shRNA were seeded at a density of 1000 or 2000 cells/well in six-well plates. Cells were exposed to the indicated doses of HU or APH at 24 h after plating and grown for 11 days at 37 °C to allow formation of colonies. Alternatively, cells were first incubated for 5 h at indicated HU dose, washed, and then plated and counted after 11 days. Colonies were stained in 0.5% (w/v) crystal violet in 25% methanol. Colonies reaching a minimum of 50 cells were counted to calculate the average plating efficiency and surviving fraction relative to control-transfected cells.

**Plasmids and RNAi treatment**. MISSION® pLKO.1-puro non-mammalian shRNA control plasmid DNA (Millipore Sigma, SHC002), human HELLS mission shRNA plasmid DNA (Millipore Sigma, TRCN0000273217 and TRCN0000273147) and mouse HELLS mission shRNA plasmid DNA (Millipore Sigma, TRCN0000039046, TRCN0000039048) were employed to generate cell lines stably expressing shRNA. RAD51 Lentifect™ Purified Lentiviral Particles (LPP-B0087-Lv203-050) were generated by GeneCopoeia. mH2A2-CT-GFP was a kind gift from Brian Chadwick and Hunt Willard. pCBASceI was a kind gift from Maria Jasin (Addgene plasmid # 26477). Plasmids were transfected into human and mouse cells using Lipofectamine 2000 (Thermo Fisher Scientific) or TurboFect Transfection Reagent (Thermo Fisher Scientific), according to the manufacturer's instructions. siRNAs (Horizon) were employed as follows, ON-TARGETplus Non-targeting pool (D-001810-10-05), siLSH (ON-TARGETplus SMART pool L-017444-01-0005), siRAD51 (ON-TARGETplus SMART pool L-003530-00-0005), simH2A1 (ON-TARGETplus SMART pool

L-01196400-0005), simH2A2 (ON-TARGETplus SMART pool L-010913-00-0005), si53BP1 (ON-TARGETplus SMART pool L-003548-00-0005), siMRE11 (ON-TARGETplus SMART pool L-009271-00-0005), siDNA2 (ON-TARGETplus SMART pool L-026431-01-0005), siEXO1 (ON-TARGETplus SMART pool L-013120-00-0005), siCTIP (ON-TARGETplus SMART pool L-011376-00-0005), siKMT5A (ON-TARGETplus SMART pool L-031917-00-0005). Oligonucleotides were transfected using Lipofectamine RNAiMAX reagent (Thermo Fisher Scientific).

**Immunoblotting.** Whole-cell extracts were carried out in RIPA lysis and extraction buffer (Thermo Fisher Scientific) supplemented with protease inhibitors (Sigma Aldrich) and phosphatase inhibitors (Thermo Fisher Scientific). Lysates were boiled with sodium dodecyl sulfate (SDS)–Laemmli sample buffer (Bio-Rad) for 10 min. $2 \times 10^5$ cells were collected for whole-cell extracts per sample. For chromatin fractionation about $1 \times 10^7$ cells were collected, washed in phosphate buffered saline (PBS), and resuspended in 1 ml buffer A (10 mM HEPES pH 7.9, 10 mM KCl, 1.5 mM MgCl$_2$, 0.34 M sucrose, 10% glycerol, 1 mM dithiothreitol, and 1× protease inhibitor cocktail). Triton X-100 was added to 0.1% (10 μl, 10% Triton X-100) and the cells were incubated for 10 min on ice. Nuclei were collected by centrifugation at $1500 \times g$ at 4 °C for 5 min. Nuclei were washed once with buffer A and then lysed for 30 min in 'No Salt' buffer B (3 mM EDTA, 0.2 mM EGTA, 1 mM DTT, and 1× protease inhibitor cocktail) on ice. Chromatin fraction was pelleted by centrifugation at $1500 \times g$ at 4 °C for 5 min. The chromatin fraction was washed once with buffer B and resuspended in SDS–Laemmli buffer followed by boiling for 10 min. Samples were resolved by SDS–PAGE and transferred to PVDF. Uncropped and unprocessed scans of all blots can be found in the Source Data file. Immunoblots were carried out using the indicated antibodies: beta-actin (Thermo Fisher Scientific, MA1-140, 1:5000), GFP (Thermo Fisher Scientific, A-6455, 1:2000), H3 (Abcam, #1791, 1:5000), H3K4me3 (Abcam, ab8580, 1:1000), H4K20me2 (Abcam, ab9052, 1:2000), MRE11 (Abcam, ab124, 1:1000), gamma-H2AX (Cell Signaling, #9718, 1:1000), marcoH2A1 (Cell Signaling, 8551S, 1:2000), marcoH2A1 (Abcam, ab183041, 1:2000), KMT5A (Cell Signaling, 2996, 1:100), Ser345-pCHK1 (Cell Signaling, 2348S, 1:500), CHK1 (Cell Signaling, 2360S, 1:500), macroH2A2 (Abcam, ab102126, 1:2000), MRE11 (Genetex, GTX70212, 1:1000), RAD51 (Millipore, PC130, 1:2500), BRCA1 (Bethyl, A300-000A, 1:1500), Ser33-pRPA2 (Bethyl, A300-246A, 1:1000), S4/S8-pRPA2 (Bethyl, A300-245A, 1:1000), 53BP1 (Novus, NB100-904, 1:2000), EXO1 (Novus, NBP2-16391, 1:500), CTIP (Santa Cruz, sc-271339, 1:100). HRP-conjugated secondary antibodies were used for signal detection by ECL western blotting substrate kit (Abcam). The secondary antibodies were anti-rabbit IgG (Abcam, ab6721, 1:5000), anti-mouse IgG (Abcam, ab6728,1:5000), anti-Rat IgG (Abcam, ab6734, 1:5000).

**RT-qPCR analysis.** The levels of siRNA-mediated knockdown for DNA2 were determined by RT-qPCR analysis. mRNAs were extracted with the RNeasy Plus Mini Kit (Qiagen) and cDNA was synthesized using the PrimeScript RT Reagent Kit with gDNA Eraser (TaKaRa). qPCR analysis was carried out with a MyiQ2 Real-Time PCR Detection System (Bio-Rad). RT-qPCR primers are listed in supplementary Table 1.

**Immunofluorescence microscopy.** For visualization of gamma-H2AX foci in lymphocytes, cells were cultured in a chamber slide (Thermo Fisher Scientific) and fixed with 4% paraformaldehyde (PFA) in PBS for 15 min at room temperature, washed three times with PBS, and permeabilized with 0.3% Triton X-100 in PBS for 10 min at room temperature. Slides were washed three times in PBS and blocked in 5% bovine serum albumin (BSA) in PBS (blocking buffer) for 40 min before incubation with the primary antibody in blocking buffer at 4 °C overnight. Slides were washed 4× 5 min in PBS with 0.1% tween-20 (PBST) followed by incubation with secondary antibody in blocking buffer at 4 °C for 1 h. Then slides were washed 4× 5 min in PBST and mounted with Vectashield antifade mounting medium with DAPI (Vector). Antibody employed for immunofluorescence as follows: gamma-H2AX (Millipore, 05-636-I, 1:100), Goat anti-Mouse Alexa Fluor 594-conjugated (Thermo Fisher Scientific, A-11005, 1:1000). Images were acquired using a Zeiss LSM780 laser scanning confocal microscope with Zen software using a ×63 objective. Images analyses were carried out with Fiji software version 2.1.0/1.53c.

**Metaphase spreads and FISH.** For sister chromatid exchanges (SCEs), 10 μM BrdU was added to the medium for two cell cycles (~48 h) in the dark before collection. Colcemid (Thermo Fisher Scientific) was added 3 h prior to harvesting at a final concentration of 0.1 μg/ml. Cells were subjected to hypotonic shock for 30 min at 37 °C in prewarmed 0.075 M KCL and fixed in ice-cold 3:1 methanol: acetic acid solution. Cells were dropped onto acetic acid-humidified slides to prepare metaphase spreads. The slides were kept at 45 °C for 18 h and were stained with a solution of Hoechst 33258 (50 μg/ml) in 2× SSC for 30 min. The slides were then mounted with McIIvaines buffer pH 7, exposed to a black light for 45 min at 50 °C. Subsequently, slides were washed with distilled water followed by staining in 4% Giemsa solution (McIIvaines buffer pH 7) for 5 min, before being washed again in distilled water, dried, and mounted. For chromosome aberrations, Colcemid (Thermo Fisher Scientific) was added 3 h prior to harvesting at a final concentration of 0.1 μg/ml. Cells were subjected to hypotonic shock for 30 min at 37 °C

in prewarmed 0.075 M KCL and fixed in ice-cold 3:1 methanol:acetic acid solution. Cells were dropped onto acetic acid-humidified slides, stained for 15 min in Giemsa-modified (Sigma Aldrich) solution (5% v/v in H$_2$O) and washed in water for 5 min. Analyses were acquired using Hyper Spectral Imaging system (Applied Spectral Imaging Inc., CA) mounted on top of an epi-fluorescence microscope (Imager Z2, Carl Zeiss). Images were analyzed using Bandview 8.2 acquisition software (Genasis, Applied Spectral Imaging Inc., CA). For the fragility FISH assay at telomeres[75], metaphase spreads were hybridized with Cy3-labeled PNA (CCCTAA)3 (0.5 μg ml/l, Panagene), washed, and mounted with prolong gold anti-fade reagent (Thermo Fisher Scientific). Images were captured using Cytovision 3.6 software (Applied Imaging) on a fluorescence microscope (Axio2; Carl Zeiss), followed by quantification of fluorescence signals of individual telomeres using Fiji software (version 2.1.0/1.53c) or the TFL-Telo software (gift from P. Lansdorp). Representative data from at least 20 metaphases of each genotype were pooled and scored using GraphPad Prism software (Version 8.0). All metaphase spreads and FISH was performed on lymphocytes.

**ChIP and real-time PCR.** For ChIP[17], U2OS cells were treated with 0.5 μM APH for 24 h, then crosslinked with a final concentration of 1.42% formaldehyde in PBS for 15 min at room temperature, followed by quenching with 125 mM glycine for 5 min at room temperature. Cells were scraped and collected by centrifugation at $2000 \times g$ for 5 min at 4 °C, lysed with IP buffer (Pierce) containing protease inhibitors according to the manufacturer instructions. The nuclear pellet was washed and resuspended in IP buffer followed by sonication to shear the chromatin. Sheared chromatin was incubated with the indicated antibodies overnight at 4 °C. For mock immunoprecipitation, we used the nonimmune IgG fraction from the same species in which the specific antibodies were generated. Immunoprecipitations were performed with Protein A/G magnetic beads (Pierce) according to the manufacturer's instructions. Eluted DNA was purified using the QIAquick PCR purification kit (Qiagen) and purified ChIP DNA was analyzed by qPCR analysis using a MyiQ2 Real-Time PCR Detection System (Bio-Rad). CHIP was carried out using the indicated antibodies (3 μg/sample): macroH2A1 (Abcam, ab37264), macroH2A2 (Abcam, ab102126), H3K4me3 (Abcam, ab8580), gamma H2AX (Abcam, ab20669,), BRCA1 (Bethyl, A300-000A), H4K20me2 (Abcam, ab9052), 53BP1 (Novus, NB100-304), Rabbit IgG (Millipore, 12-370). For ChIP PCR quantification, the standard comparative cycle threshold method was used to measure the amount of DNA. CHIP-qPCR primers are listed in Supplementary Table 1.

**ChIP seq-analysis.** ChIP seq-analysis of three pairs of macroH2A1 and three pairs of macroH2A2 samples and their respective input samples derived from three *Lsh*−/− murine embryonic fibroblast (MEF) cell lines (LSH-KO) and three *Lsh*+/+ controls (WT) has been previously reported[17]. Normalization to input control, tabulation of macroH2A (macroH2A1 and macroH2A2) enrichment in 5 kb tiles across the genome, peak call and domain call has been previously described[17]. The genome coordinates of human CFSs[49] were converted to the MM10 genome to identify syntenic regions using the online UCSC genome browser tool LiftOver (http://www.genome.ucsc.edu/cgi-bin/hgLiftOver). Early replicating fragile sites have been described previously[50]. The coordinates of repeat sequences (rmsk track) are available from UCSC (https://hgdownload.soe.ucsc.edu/goldenPath/mm10/database/). The repeat sequences, CFSs, and early replicating fragile sites were tabulated in 5 kb bins of the MM10 genome resulting in specific frequency of each repeat type across the genome based on 5 kb bins. The genomic regions containing fragile sites were intersected with macroH2A peaks or macroH2A domains and intersecting peak or domain counts were reported.

**DNA fiber analysis.** For the DNA fiber assay[36], cells were seeded in six wells plate at about 20% confluency on the previous day. Cells were first incubated with 50 μM CIdU for 20 min at 37 °C followed by three times wash with 37 °C PBS, and then incubated with 250 μM IdU for 20 min at 37 °C. The cells were washed with 37 °C PBS for three times before incubation with 4 mM HU for 5 h. Cells were harvested and resuspended in cold PBS to $2 \times 10^5$ cells/ml and 2.5 μl of resuspended cells were mixed directly with 7.5 μl of lysis buffer (200 mM Tris–HCl pH 7.4, 50 mM EDTA, 0.5% SDS) on slides. The slides were left horizontally for 8 min at room temperature and then tilted to allow drops to run slowly down the slides, followed by air dry and fixation in 3:1 methanol:acetic acid solution overnight at 4 °C. Slides were rehydrated in PBS and DNA denatured in 2.5 M HCl. Slides were washed several times in PBS until pH was back to 7–7.5. Afterwards slides were blocked in 2% BSA, 0.1% Tween 20 in PBS, and immuno-fluorescence staining performed. Slides were mounted with prolong gold antifade mounting medium (Thermo Fisher Scientific). Antibodies employed for immunofluorescence were as follows: anti-CIdU (Abcam, ab6326, 1:300), anti-IdU (BD bioscience, #347580, 1:500), goat anti-mouse Alexa Fluor 488- conjugated (Thermo Fisher Scientific, A-11001, 1:1000), donkey anti-rat Cy3-conjugated (Jackson Immunoresearch, 712-166-153, 1:1000). Images were acquired using Zeiss Axio Imager M2 or Zeiss Axio Imager D2 microscope with Zen2 software (blue edition) using a ×63 objective. Image analyses were carried out with Fiji software (version 2.1.0/1.53c). For fork restart analysis, the labeling was carried out as indicated in the figure and all other procedures were similar as above.

**Proximity ligation assay**. For the association of proteins to newly synthesized DNA using EdU labeling and the proximity ligation assay (PLA)[52]. U2OS cells were cultured in chamber slides and labeled with 10 µM EdU for 20 min followed by treatment with 4 mM HU for 5 h. Cells were fixed with 4% PFA in PBS for 15 min at room temperature, washed twice with PBS, and permeabilized with 0.3% Triton X100 in PBS for 10 min at room temperature. Click-iT reaction (Thermo Fisher Scientific) was carried out to conjugate biotin-azide to the newly synthesized EdU according to the manufacturer's instruction using 1 µM biotin-azide for 30 min. Slides were washed twice before blocking with Duolink blocking solution at 37 °C for 1 h. Following primary antibodies incubation at 4 °C for overnight, slides were washed twice and incubated with secondary antibodies followed by ligation and amplification reaction according to the protocol from Duolink in situ detection reagents red kit (Sigma). EdU was stained by antibody with Alexa Fluor 488 or Alexa Fluor 647 as indicated in figures followed by DAPI staining. The slides were mounted with antifade mounting medium (Thermo Fisher Scientific). Antibodies employed for PLA were as follows: BRCA1 (Sant Cruz, sc-6954, 1:200), H3 (Abcam, #1791, 1:1000), H3K4me3 (Abcam, ab8580, 1:1000), H4K20me2 (Abcam, ab9052, 1:2000), mH2A1 (Cell Signaling, 8551S, 1:500), mH2A2 (Abcam, ab102126, 1:200), RAD51 (Millipore, PC130, 1:250), Ser33-pRPA2 (Bethyl, A300-246A, 1:200), 53BP1 (Novus, NB100-904, 1:500), Donkey anti-Rabbit Alexa Fluor 488-conjugated (Thermo Fisher Scientific, A-21206, 1:1000), Goat anti-Mouse Alexa Fluor 488-conjugated (Thermo Fisher Scientific, A-11001, 1:1000). Images were acquired using Zeiss LSM780 laser scanning confocal microscope with Zen software using a ×40 objective. Images analyses were carried out with Fiji software (version 2.1.0/1.53c).

**iPOND**. For isolation of proteins on nascent DNA (iPOND)[53], a total of $1 \times 10^8$ U2OS cells were labeled with 10 µM EdU for 10 min. After EdU incorporation, the cultured cells were incubated in 4 mM of hydroxyurea for 5 h. All samples were fixed with 1% formaldehyde for 20 min at room temperature, followed by adding 0.125 M glycine for 10 min to quench the formaldehyde. Cells were collected by scraping on ice, transferred into 50 ml conical tubes, centrifuged at 900 × g at 4 °C for 5 min, washed three times with cold PBS, and frozen at 80 °C. For click reactions, cells were resuspended in permeabilization buffer at a concentration of $1 \times 10^7$ cells/ml for 30 min at room temperature. After a wash with cold 0.5% BSA-PBS and PBS twice by centrifuging cells at 900 × g for 5 min, the click chemistry reaction was performed to conjugate biotin to the EdU-labeled DNA in the buffer containing a final concentration of 10 mM sodium ascorbate, 2 mM CuSO4, 10 µM biotin-azide. The "NO click" samples were treated in reaction buffer without the biotin-azide. After rotating reactions tubes at room temperature for 2 hrs, cells were washed with cold 0.5% BSA–PBS and PBS twice at 900 × g for 5 min. Cells were lysed in a buffer containing aprotinin and leupeptin and sonicated using Bioruptor300 (Diagenode). After samples were centrifuged for 10 min at 16,100 × g, the supernatant was filtered through a 90-micron nylon mesh. Streptavidin agarose beads were used to capture the biotin-conjugated DNA–protein complexes by rotating at 4 °C overnight. Captured complexes were washed extensively using SDS and high-salt wash buffers at least twice. Purified replication fork proteins and input samples were boiled in SDS sample buffer for 25 min before separation by standard SDS–PAGE.

**Neutral comet assay**. For the neutral comet assay[34], slides were precoated with 1% low-gelling-temperature agarose (Sigma). Harvested cells were prepared as single-cell solutions to about $2 \times 10^4$ cells/ml in PBS lacking divalent cations and mixed with 1% low-gelling-temperature agarose at 40 °C. The cell suspension was pipetted onto the agarose-covered surface of the precoated slide avoiding bubbles. After agarose was gelled, slides were gently submerged in a covered dish with neutral lysis solution (N1) containing 2% sarkosyl, 0.5 M EDTA-Na$_2$, 0.5 mg/ml proteinase (pH 8.0) at 4 °C. The dishes were placed in an incubator at 37 °C for 18–20 h in the dark. Slides were removed from dishes and washed three times in rinse buffer (N2) containing 90 mM Tris buffer, 90 mM boric acid, and 2 mM EDTA-Na$_2$ (pH 8.5). Slides were then submerged in fresh N2 solution in an electrophoresis chamber and electrophoresis were conducted for 25 min at 0.6 V/cm. Following neutralization in distilled water, slides were stained with 2.5 µg/ml propidium iodide in distilled water for 20 min. Slides were washed several times to remove excess stain and the images were acquired using an Olympus IX70 Fluorescence Microscope with MagnaFire SP camera using a ×20 objective. At least 50 comet images from each slide were analyzed and tail length were measured by Fiji software (version 2.1.0/1.53c).

**DR-GFP reporter assay**. MCF10A DR-GFP reporter cells were transfected by indicated shRNA and stable cells were selected as described above. The coding sequence for the SceI enzyme (pCBASceI) was cloned into lentiviral vector by GeneCopoeia and lentivirus was used to transfect reporter cells twice with an interval of 3 days. The cells were harvested for flow cytometry (BD LSR II) and immunoblotting analysis. About $5 \times 10^5$ cells were analyzed per experiment for flow cytometry. The percentage of GFP-positive cells was normalized to SceI enzyme expression based on immunoblotting analysis. Flow cytometry data were assessed using BD FACSDiva software (version 6.1.3) and used the same gating strategies. Supplementary figure to graphically account for FACS sequential gating

strategies has been provided. Gray values of immunoblotting data were assessed using Fiji software (version 2.1.0/1.53c).

**Statistics and reproducibility**. All ChIP-qPCR and RT-qPCR experiments were repeated at least three times as independent experiments and are presented as mean ± SD. All immunoblots were repeated at least two times in independent experiments and led to similar results. Micrographs are representative of a minimum of eight images taken from at least three independent experiments. For Fig. 1a, d; 2b–d, f, g; 3c, f; 4a, b, f–h; 5a, c, e, f; 6a, d, f; 7a–d, f; 8a–c, e–g and Supplementary Figs. 4d; 5e, f; 12b, c, at least three independent experiments were performed with similar results and representative results are shown. Statistical analysis was performed using GraphPad Prism software (Version 8.0). Student's two-tailed $t$ test, two-tailed Mann–Whitney test and two-tailed one-proportion $z$ test were used to determine statistical significance as indicated in the figure legends. In all cases: n.s. $p > 0.05$; $*p < 0.05$; $**p < 0.01$; $***p < 0.001$; $****p < 0.0001$. All figures were prepared by Fiji software (version 2.1.0/1.53c) and Microsoft Power-Point (version 16.47).

**Reporting summary**. Further information on research design is available in the Nature Research Reporting Summary linked to this article.

## Data availability
Data supporting the findings of this study are available within the paper and its Supplementary Information files and are available from the corresponding author upon reasonable request. MacroH2A1-seq data and macroH2A2-seq data were obtained from our published work[17] and can be found in Gene Expression Omnibus, GEO accession number: "GSE142082"). Source data are provided with this paper.

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

## Acknowledgements

We wish to thank Karen Saylor from the FNLCR LASP Animal Research and Technical Support Staff. We thank Dr. Maria Jasin for the kind gift MCF10A DR-GFP cells and Dr. Peter Lansdorp for sharing the TFL-Telo software. This project has been funded in whole or in part with federal funds from the National Cancer Institute, National Institutes of Health, under contract HHSN26120080001E. The content of this publication does not necessarily reflect the views or policies of the Department of Health and Human Services, nor does mention of trade names, commercial products, or organizations imply endorsement by the U.S. Government. This research was supported by the Intramural Research Program of the NIH, National Cancer Institute, Center for Cancer Research. Frederick National Laboratory for Cancer Research is accredited by AAALAC International and follows the Public Health Service Policy for the Care and Use of Laboratory Animals. Animal care was provided in accordance with the procedures outlined in the "Guide for Care and Use of Laboratory Animals" (National Research Council; 1996; National Academy Press, Washington, DC).

## Author contributions

X.X. and K.M. conceived and designed experiments. X.X., K.N., Y.H., J.R., C.S., S.B., X.D. conducted experiments. Y.L., M.I.A., S.K.S. provided crucial reagents, expertise, and guidance. R.F. performed bioinformatic analysis. X.X., K.N., and K.M. analyzed data, and X.X. and K.M. wrote the manuscript.

## Funding

## Competing interests

The authors declare no competing interests.
