## [Peer Review File · Nature Communications]

REVIEWER COMMENTS

Reviewer #1 (Remarks to the Author):

Xu et al present interesting new evidence indicating that the chromatin remodeling factor LSH is required for genome stability in somatic cells.

Depletion of LSH induced degradation of nascent DNA at stalled replication forks and DNA damage. LSH is required for incorporation of macroH2A and its effect on protection of the replication fork is mediated by MacroH2A as indicated by rescuing the phenotype following over-expression of macroH2A in LSH deficient cells.

The authors present novel mechanistic evidence indicating that loss of LSH induces deficient RAD51 loading due to abnormal BRCA1 and 53BP1 accumulation at stalled replication forks in somatic cells. In turn, defects in the BRCA1/53BP1 balance were associated with abnormal H4K20 methylation.

This is an interesting manuscript that provides novel mechanistic insight into the function of LSH on genome stability. The results are convincing and support most of the conclusions reached by the authors. However, a few statements need to be revised for clarity.

Specific comments to the authors:

How specific is the Mirin inhibitor? Please provide information on the spectrum of this inhibitor at the doses used here.

The conclusion that LSH prevents nucleolytic degradation mediated by MRE11 nuclease are based only on an inhibitor experiment. Additional evidence is required to substantiate this conclusion or tone down the statement.

In addition to the use of the RS-1 inhibitor, the authors should directly over-express RAD51 to strengthen the evidence presented here.

Page 9: Additional information is needed in the text as to how was macroH2A1 depleted

Page 18: The last sentence before the subheading is not clear. Please clarify.

Check for the correct nomenclature (See:H4K20m0 in page 19).

The fragile telomere defects in Figure 1 D, are difficult to appreciate. Please include micrographs at higher resolution and with larger insets.

What kind of chromosome aberrations are depicted in Figure 1 C?

How long where the cells exposed to RS-1 and what is the viability of these cells after treatment?

The comet assays in Figure 5 E show minimal differences in tail length between treatment groups.

Reviewer #2 (Remarks to the Author):

Here the authors describe the role in the maintenance of genomic stability of LSH, a protein encoded by a gene mutated in the IMMUNODEFICIENCY-CENTROMERIC INSTABILITY-FACIAL ANOMALIES SYNDROME 4 (ICF4).

Overall this is an interesting and well-crafted paper, with clear experiments that provide new insights on the role and the molecular mechanisms underlying LSH dependent protection of nascent DNA at

replication forks under stressful conditions. This said some aspects and claims of the work need to be revised as not fully supported. In particular, the proposal that LSH avoids genomic instability via RAD51 filament formation is not fully substantiated. Furthermore, the nuclease(s) responsible for DNA degradation in LSH deficient cells should be better characterized. Finally, the links between fork protection issues and centromeric defects observed in ICF4 cells should be discussed.

Major critiques concerns:

1. The authors claim that LSH deficient cells are more susceptible to replication stress, but this was not clearly shown. To prove this, I suggest performing a clonogenic assay in WT and LSH deficient cells exposed to different concentrations of HU and/or aphidicolin.
2. The authors demonstrated that in LSH deficient cells under replication stress nascent DNA is degraded. This degradation was partially rescued by the chemical inhibition of MRE11. It is evident that a nucleolytic process is involved. However, it seems that MRE11 is not the only nuclease responsible for this phenotype since the inhibition could not fully rescue the degradation. In order to have a better picture of the mechanism(s) involved, I suggest repeating the experiment depicted in figure 2F using siRNAs against the main factors involved in nascent DNA degradation (DNA2, EXO1, CTIP, MRE11, ... etc) to better identify the nuclease(s) responsible of the phenotype.
3. The authors suggest that drugs that stabilize RAD51 filaments may serve as a novel target for ICF4 syndrome. They show that RAD51 stabilization reduces DNA damage. I suggest testing whether RS-1 also rescues the chromosomal aberrations in LSH deficient cells.
4. The links with centromere defects observed in LSH should be highlighted. In ICF4 cells centromeres are hypomethylated. Centromeres have been shown to harbor secondary structures that might lead to spontaneous replication stress (Aze et al NCB 2016). It is possible that in hypomethylated centromeres with defective chromatin such secondary structures are subjected to nuclease mediated processing (Li Z. et al EMBOJ 2018), which might trigger nascent DNA degradation. These references could be cited and discussed to possibly link LSH defects to the observed centromeres in ICF4 syndrome.

Minor points.

-On page 9 there is a typo in the line: "First we overexpressed macroH2A..."

-On page 17 there is a typo before the line: "Our model suggest that LSH..."

-The original reference showing direct RAD51 nucleofilament mediated protection of nascent DNA is missing and should be included (Kolinjivadi et al, Mol Cell 2017).

Reviewer #3 (Remarks to the Author):

The manuscript by Xu et al. documents a new role for the chromatin remodeller LSH/HELLS in regulating the stability of DNA replication forks. LSH-deficient cells experience genome instability in the form of DNA replication stress. Characterization of the nature of the replication stress uncovers a potential role for LSH in regulating the incorporation of Rad51 on chromatin to protect stalled replications forks from resulting in DSBs. Their observations show a dysregulation of fork stabilizing factors that appears to be mediated through the lack of incorporation of macroH2A2 genome-wide. The authors propose that loss of LSH promotes fork degradation by creating a chromatin environment favorable to MRE11 degradation. Overall this is an interesting study which provides new insight to the roles of chromatin regulators in replication fork protection and restart. Below I suggest several ways in which the manuscript can be improved through additional experiments, revised text and clarifications.

Major comments:

1. CHIP-qPCR data is convincing but looking at the distribution of mH2A1/2 and yH2AX in LSH KOs would be interesting and might uncovered specific regions of the genome that are altered by LSH KO. Some of this data may be available in the companion manuscript Ni et al. that the authors refer to. It would be nice to have more analysis of this data in the present manuscript. Do mH2A changes show any preference for fragile sites? Are other chromatin marks (e.g. those highlighted in Figure 7) changed a specific sites where mH2A incorporation is changed? A deep consideration of the chromatin state of difficult to replicate regions would help us understand more about the proposed direct or indirect role of

histone variants and modifications controlled by LSH at replication forks.

2. a. The 'abnormal' chromatin environment proposed in Figure 7 is a key part of the model linking LSH to mH2A to histone modifications and could be enhanced with more direct tests in a number of ways. First, it would be more convincing to show that siRNA depletion of macroH2A also decreases H3K4me3 and H4K20me2.

2b. CHIP data of the H3K4me3 and H4K20me2 at the fragile sites induced in Fig. 6C would directly link the local changes in these PTMs to fragile sites influenced by LSH providing a nice direct model in the absence of genome-wide data.

2c. Indeed, directly modulating SET7/8 with siRNA could be shown to reduce H4K20me2-EdU PLA foci, and concomitantly reduce 53BP1-EdU and increase BRCA1-EdU PLA. Thus further linking the authors' model with direct data.

2c. Finally, I also think the authors should draw a schematic of their model linking LSH to mH2A depletion to altered histone PTMs and defective recruitment of DNA repair proteins to stalled forks.

3. Given that the authors state that loss of BRCA1 occupancy in LSH deficient cells may contribute to failure of RAD51 loading, the authors should show a western blot confirming that BRCA1 protein levels are not affected.

4. Aphidicolin treatments are shown in many experiments as a control. However, remarkably the rationale for this treatment and the interpretation of the Aphidicolin results are never discussed in the main text. APH is only mentioned in the figure legends. The authors need to describe what this treatment means to the readers. In some cases there is confusion about whether the authors are referring to HU or aphidicolin. The use of APH to induce fragile sites needs to be described.

5. The Rad51 western blot in Figure 5B needs to be quantified and compared to the loading control. The effects of HU are very subtle and it's hard to understand how LSH loss + RS-1 leads to more Rad51 than WT + RS-1. Repetition and quantification would help to clarify the variance in this assay.

6. In Figure 6 adding CHIP data of BRCA1 occupancy at fragile sites in LSH-deficient cells with or without si53BP1 could help to strengthen the BRCA1-53BP1 competition model.

Minor comments:

- The combing experiments are only clearly outlined in Figure 2. The authors must make sure that the method of CldU/IdU ratio calculations in Figure 3, 5, and 6 are clearly described in the legend of a schematic. I assume they are all based on the method described for Fig. 2A so this could be indicated easily.
- CHIP-qPCR of mH2A2 enrichment at CFSs (supplemental 4a and figure 3d): the values are quite different despite being basically the same experiment (although one is siRNA and the other is shRNA).
- Figure 6: they suggest proper activation of the DNA damage repair cascade just based on RPA phosphorylation. It would be nice to show by western activation of ATR, and other DNA damage markers.
- They begin with a mice model looking at lymphocytes and then move on to U2OS (bone cancer line). Why move to U2OS instead of a lymphocyte line?
- Figure S2 – the authors should also show no change in CldU tract and reduction in IdU tract like they did for U2OS. It is convincing for U2OS since the CldU tract did not change. But without that info again, a reduction in IdU/CldU could be interpreted in different ways.
- Is macroH2A also playing a role in modulating RPA phosphorylation?
- Presenting a bit more of the background literature on macroH2A incorporation by LSH would improve the introduction and/or discussion. What other factors are involved in macroH2A incorporation? What defines specificity of mH2A1 vs. mH2A2 and is it possible to create LSH mutants that separate these functions?
- I suggest relabeling the legend in Figure 5C to say LSH-KO instead of just KO.
- Some signal seems to have been cut off in the edges of Figure 4E, cleaning up the image and/or showing the whole blot would be good.

REVIEWER COMMENTS

Reviewer #1 (Remarks to the Author):

Xu et al present interesting new evidence indicating that the chromatin remodeling factor LSH is required for genome stability in somatic cells.

Depletion of LSH induced degradation of nascent DNA at stalled replication forks and DNA damage. LSH is required for incorporation of macroH2A and its effect on protection of the replication fork is mediated by MacroH2A as indicated by rescuing the phenotype following over-expression of macroH2A in LSH deficient cells.

The authors present novel mechanistic evidence indicating that loss of LSH induces deficient RAD51 loading due to abnormal BRCA1 and 53BP1 accumulation at stalled replication forks in somatic cells. In turn, defects in the BRCA1/53BP1 balance were associated with abnormal H4K20 methylation.

This is an interesting manuscript that provides novel mechanistic insight into the function of LSH on genome stability. The results are convincing and support most of the conclusions reached by the authors. However, a few statements need to be revised for clarity.

Specific comments to the authors:

How specific is the Mirin inhibitor? Please provide information on the spectrum of this inhibitor at the doses used here.

R: Mirin has been validated as pharmacological inhibitor of MRE11 exonuclease activity and a range of concentrations from 25 to 100 μ M were found inhibitory for homologous recombination in mammalian cells (Dupre, Boyer-Chatenet et al. 2008, Shibata, Moiani et al. 2014).

We used 50 μ M of Mirin, a concentration which had been used in previous studies (Her, Ray et al. 2018; Lemaçon, Jackson et al. 2017).

This information has been added on page 7,8 and in figure legend 2F.

However, to validate the specificity we have now new data in the revised manuscript (below) that further corroborates our findings on MRE11.

The conclusion that LSH prevents nucleolytic degradation mediated by MRE11 nuclease are based only on an inhibitor experiment. Additional evidence is required to substantiate this conclusion or tone down the statement.

R: We have now used siRNA to deplete MRE11 and found that MRE11 is required for degradation of nascent DNA in LSH deficient cells consistent with the data generated by mirin. The new data is presented in Fig 2G and Fig S6B and discussed on page 8.

In addition to the use of the RS-1 inhibitor, the authors should directly over-express RAD51 to strengthen the evidence presented here.

R: We have now overexpressed RAD51 and observed that overexpression of RAD51 can partially rescue the detrimental effect of Lsh deficiency suggesting that nascent DNA degradation in the absence of LSH is in part mediated by impaired RAD51 filament formation. The data is presented in Fig 4E, H and discussed on page 12.

Page 9: Additional information is needed in the text as to how was macroH2A1 depleted
macroH2A

R: MacroH2A1 and macroH2A2 were depleted by RNA interference using small interfering RNA (siRNA). This information is now added on page 10 and in the legend of Fig.S8D.

Page 18: The last sentence before the subheading is not clear. Please clarify.

R: We clarified the sentence.

Check for the correct nomenclature (See:H4K20m0 in page 19).

R: We checked the correct nomenclature and corrected errors.

The fragile telomere defects in Figure 1 D, are difficult to appreciate. Please include micrographs at higher resolution and with larger insets.

R: We include now better micrographs with higher resolution and less background in Fig 1D and a blowup version of the same image in Fig S2.

What kind of chromosome aberrations are depicted in Figure 1 C?

R: The most frequent chromosomal aberrations that were observed are breaks, gaps, arm losses and dicentric translocations. This information is now added to the legend of Fig 1C.

How long where the cells exposed to RS-1 and what is the viability of these cells after treatment?

R: The cells were exposed for 5hrs and we have now examined viability after RS-1 treatment under the experimental conditions and found no effect on viability. This data is shown in Fig S13A and reported on page 13.

The comet assays in Figure 5 E show minimal differences in tail length between treatment groups.

R: We agree that the quality of the image does not reflect the statistical significant difference between the groups, and we hope that we have now chosen a more representative image.

Reviewer #2 (Remarks to the Author):

Here the authors describe the role in the maintenance of genomic stability of LSH, a protein encoded by a gene mutated in the IMMUNODEFICIENCY-CENTROMERIC INSTABILITY-FACIAL ANOMALIES SYNDROME 4 (ICF4).

Overall this is an interesting and well-crafted paper, with clear experiments that provide new insights on the role and the molecular mechanisms underlying LSH dependent protection of nascent DNA at replication forks under stressful conditions. This said some aspects and claims of the work need to be revised as not fully supported. In particular, the proposal that LSH avoids genomic instability via RAD51 filament formation is not fully substantiated. Furthermore, the nuclease(s) responsible for DNA degradation in LSH deficient cells should be better characterized. Finally, the links between fork protection issues and centromeric defects observed in ICF4 cells should be discussed.

Major critiques concerns:

1. The authors claim that LSH deficient cells are more susceptible to replication stress, but this was not clearly shown. To prove this, I suggest performing a clonogenic assay in WT and LSH deficient cells exposed to different concentrations of HU and/or aphidicolin.

R: We have now conducted clonogenic assays in LSH deficient U2OS cells and controls exposed to different concentrations of HU or aphidicolin in different time frames and show susceptible to replication stress upon LSH depletion. The data is presented in Fig S3A-D and discussed on page 6.

2. The authors demonstrated that in LSH deficient cells under replication stress nascent DNA is degraded. This degradation was partially rescued by the chemical inhibition of MRE11. It is evident that a nucleolytic process is involved. However, it seems that MRE11 is not the only nuclease responsible for this phenotype since the inhibition could not fully rescue the degradation. In order to have a better picture of the mechanism(s) involved, I suggest repeating the experiment depicted in figure 2F using siRNAs against the main factors involved in nascent DNA degradation (DNA2, EXO1, CTIP, MRE11, ... etc) to better identify the nuclease(s) responsible of the phenotype.

R: We have now performed knockdown of the suggested selected nucleases. Depletion of MRE11 or depletion of EXO1 was able to partially restore nascent DNA protection, whereas DNA2 depletion did not restore and CTIP depletion slightly aggravated the fork degradation consistent with previous studies (Lemaçon, Jackson et al. 2017; Przetocka, Porro et al. 2018). The results are presented in Fig 2G and Fig S6B,C and discussed on page 8.

3. The authors suggest that drugs that stabilize RAD51 filaments may serve as a novel target for ICF4 syndrome. They show that RAD51 stabilization reduces DNA damage. I suggest testing whether RS-1 also rescues the chromosomal aberrations in LSH deficient cells.

R: We have now tested RS-1 in cultures of LSH deficient lymphocytes and found a small but significant decrease on chromosomal aberrations. The data is presented in Fig 5F and discussed on page 13.

4. The links with centromere defects observed in LSH should be highlighted. In ICF4 cells

centromeres are hypomethylated. Centromeres have been shown to harbor secondary structures that might lead to spontaneous replication stress (Aze et al NCB 2016). It is possible that in hypomethylated centromeres with defective chromatin such secondary structures are subjected to nuclease mediated processing (Li Z. et al EMBOJ 2018), which might trigger nascent DNA degradation. These references could be cited and discussed to possibly link LSH defects to the observed centromeres in ICF4 syndrome.

R: This link is now included in the discussion section on page 20.

Minor points.

-On page 9 there is a typo in the line: “First we overexpressed macroH2A...”

R: We corrected the typo.

-On page 17 there is a typo before the line: “Our model suggest that LSH..”

R: We corrected the error.

-The original reference showing direct RAD51 nucleofilament mediated protection of nascent DNA is missing and should be included (Kolinjivadi et al, Mol Cell 2017).

R: We added the reference on page 3 and 10.

Reviewer #3 (Remarks to the Author):

The manuscript by Xu et al. documents a new role for the chromatin remodeller LSH/HELLS in regulating the stability of DNA replication forks. LSH-deficient cells experience genome instability in the form of DNA replication stress. Characterization of the nature of the replication stress uncovers a potential role for LSH in regulating the incorporation of Rad51 on chromatin to protect stalled replication forks from resulting in DSBs. Their observations show a dysregulation of fork stabilizing factors that appears to be mediated through the lack of incorporation of macroH2A2 genome-wide. The authors propose that loss of LSH promotes fork degradation by creating a chromatin environment favorable to MRE11 degradation. Overall this is an interesting study which provides new insight to the roles of chromatin regulators in replication fork protection and restart. Below I suggest several ways in which the manuscript can be improved through additional experiments, revised text and clarifications.

Major comments:

1. CHIP-qPCR data is convincing but looking at the distribution of mH2A1/2 and γ H2AX in LSH KO cells would be interesting and might uncover specific regions of the genome that are altered by LSH KO. Some of this data may be available in the companion manuscript Ni et al. that the authors refer to. It would be nice to have more analysis of this data in the present manuscript. Do mH2A changes show any preference for fragile sites? Are other chromatin marks (e.g. those highlighted in Figure 7) changed at specific sites where mH2A incorporation is changed? A deep consideration of the chromatin state of difficult to replicate regions would help us understand more about the proposed direct or indirect role of histone variants and modifications controlled by LSH at replication forks.

R: We performed now an analysis of our previously published ChIP-seq data on macroH2A differences with respect to sites prone to replication stress.

We found that common fragile sites and early replication fragile sites show significant decreases in macroH2A enrichment in LSH deficient cells compared to wild type cells, making them a potential target of replication stress in the absence of LSH function.

In addition, we found an association of the magnitude of macroH2A differences, certain repeat sequences and the presence of early fragile sites such that the genomic regions with the greatest loss of macroH2A in LSH deficient cells showed a greater frequency of early fragile sites and certain types of repeats.

Altogether, the data suggest that sites prone to replication stalling are significantly affected by macroH2A loss in LSH deficient cells making them potentially more susceptible to replication stress.

Further ChIP analysis confirms changes in H4K20me2, and H3K4me3 at common fragile sites (see below). Together with BRCA1 changes (see below), 53BP1 changes and macroH2A changes our data is consistent with a model that underlines the importance of the chromatin environment to the resolution of DNA damage caused by replication stress in LSH deficient cells.

The data is presented in Fig S7 and discussed on page 9.

2. a. The 'abnormal' chromatin environment proposed in Figure 7 is a key part of the model linking LSH to mH2A to histone modifications and could be enhanced with more direct tests in a

number of ways. First, it would be more convincing to show that siRNA depletion of macroH2A also decreases H3K4me3 and H4K20me2.

R: Now we have conducted the suggested PLA experiment and show that macroH2A depletion alters H3K4me3, H4K20me2 and also phosphorylated RPA-ser33 at the stalled replication fork in the same way LSH depletion alters these marks. This further supports our notion that macroH2A depletion mimics LSH knockdown. The data is presented in Fig 7F and Fig 8C and discussed on page 15,16.

2b. ChIP data of the H3K4me3 and H4K20me2 at the fragile sites induced in Fig. 6C would directly link the local changes in these PTMs to fragile sites influenced by LSH providing a nice direct model in the absence of genome-wide data.

R: Using ChIPs we now show that H3K4me3 and H4K20me2 are increased at fragile sites in LSH deficient cells compared to controls. The data is shown in Fig 7E and Fig 8D for CFSs and Fig S16A, B for satellites and discussed on pages 15,16.

2c. Indeed, directly modulation SET7/8 with siRNA could be shown to reduce H4K20me2-EdU PLA foci, and concomitantly reduce 53BP1-EdU and increase BRCA1-EdU PLA. Thus further linking the authors model with direct data.

R: Now we have reduced SET7/8 (also known as KMT5A) by siRNA in LSH knockdown cells, and found that it reduces H4K20me2-EdU PLA foci and 53BP1-EdU foci, while BRCA1-EdU PLA foci are increased.

The data further supports the notion that 53BP1 and BRCA1 are in a balance at the fork and is presented in Fig 8E, F, G and Fig S16C and discussed on page 16.

2c. Finally, I also think the authors should draw a schematic of their model linking LSH to mH2A depletion to altered histone PTMs and defective recruitment of DNA repair proteins to stalled forks.

R: We have now added a schematic of our model in Fig S17.

3. Given that the authors state that loss of BRCA1 occupancy in LSH deficient cells may contribute to failure of RAD51 loading, the authors should show a western blot confirming that BRCA1 protein levels are not affected.

R: Now we provide a western analysis which confirms that BRCA1 protein levels are not affected in LSH deficient cells as shown in Fig S14A and mentioned on page 14.

4. Aphidicolin treatments are shown in many experiments as a control. However, remarkably the rationale for this treatment and the interpretation of the Aphidicolin results are never discussed in the main text. APH is only mentioned in the figure legends. The authors need to describe what this treatment means to the readers. In some cases there is confusion about whether the authors are referring to HU or aphidicolin. The use of APH to induce fragile sites needs to be described.

R: The use of aphidicolin to induce fragile sites is now described on page 6 and pointed out throughout the text.

5. The RAD51 western blot in Figure 5B needs to be quantified and compared to the loading control. The effects of HU are very subtle and its hard to understand how LSH loss + RS-1 leads

to more Rad51 than WT + RS-1. Repetition and quantification would help to clarify the variance in this assay.

R: We have repeated the western analysis and quantified and found some recovery on RAD51 chromatin association in LSH deficient cells under RS-1 treatment. The data is shown in Fig 5B and presented on page 13.

6. In Figure 6 adding ChIP data of BRCA1 occupancy at fragile sites in LSH-deficient cells with or without si53BP1 could help to strengthen the BRCA1-53BP1 competition model.

R: We provide now ChIP data on BRCA1 and find greatly reduced BRCA1 enrichment at CFSs. Moreover, depletion of 53BP1 leads to a partial recovery of BRCA1 association which further strengthens the model. The data is presented in Fig 6B and Fig 6G and discussed on pages 15, 16.

Minor comments:

- The combing experiments are only clearly outlined in Figure 2. The authors must make sure that the method of CldU/IdU ratio calculations in Figure 3, 5, and 6 are clearly described in the legend of a schematic. I assume they are all based on the method described for Fig. 2A so this could be indicated easily.

R: A comment is added to the respective Fig legends that all IdU/CldU ratio calculations are done as outlined for Fig2A.

- ChIP-qPCR of mH2A2 enrichment at CFSs (supplemental 4a and figure 3d): the values are quite different despite being basically the same experiment (although one is siRNA and the other is shRNA).

R: For experiments using shRNA treated cells we collected around 1×10^7 cells, but only around 2×10^6 cells were used for siRNA treated samples. This difference in cell number may influence the sonication condition which in turn can lead to slight differences of DNA fragmentation and immunoprecipitation efficiency. Importantly, the same cell numbers and conditions were used for control and experimental group for each type of experiment. The data of supplemental 4a has been moved to Fig S8A.

- Figure 6: they suggest proper activation of the DNA damage repair cascade just based on RPA phosphorylation. It would be nice to show by western activation of ATR, and other DNA damage markers.

R: Now we show activation of ATR by western examining Ser345 phosphorylation of CHK1 in the presence or absence of a specific ATR inhibitor which is shown in Fig S11B and commented on page 11.

- They begin with a mice model looking at lymphocytes and then move on to U2OS (bone cancer line). Why move to U2OS instead of a lymphocyte line?

R: We wanted to demonstrate that the nascent DNA defect is present in different cell types and species. In addition, replication stress and the occurrence of common fragile sites is well documented in human cells. Also, U2OS cells can be easily transfected.

• Figure S2 – the authors should also show no change in CldU tract and reduction in IdU tract like they did for U2OS. It is convincing for U2OS since the CldU tract did not change. But without that info again, a reduction in IdU/CldU could be interpreted in different ways.

R: The CldU tract of the collected data used for determining the IdU/CldU ratio was slightly increased, possibly indicating differences in embryonic cells.

However, given that the experiments were performed under identical conditions for shControl and shLSH, given that both CldU and IdU incorporation time was identical, and given that there was a significant reduction in IdU tracks relative to CldU tracts, despite CldU tracts being slightly longer, the reduced ratio suggests degradation of the nascent DNA in LSH deficient ES cells.

This method of normalizing IdU tract length to CldU tract length (as IdU/CldU ratio) is frequently used to measure resection of nascent DNA and has been employed in other studies without applying further normalization techniques (Higgs, Reynolds et al. 2015; Ding, Ray Chaudhuri et al. 2016; Ray Chaudhuri, Callen et al. 2016). The data is presented in Fig S5F and commented on page 7.

• Is macroH2A also playing a role in modulating RPA phosphorylation?

R: We show here that depletion of macroH2A mimics LSH depletion: it results also in an increase of Ser 33 phosphorylated RPA at the stalled replication fork concomitant with a decrease of RAD51 (Fig 4F, G). This increase may be in part due to a failure to replace RPA with RAD51. The data is shown in Fig 4G and commented on page 12.

• Presenting a bit more of the background literature on macroH2A incorporation by LSH would improve the introduction and/or discussion. What other factors are involved in macroH2A incorporation? What defines specificity of mH2A1 vs. mH2A2 and is it possible to create LSH mutants that separate these functions?

R: We added background literature on pages 4,18, and 19.

• I suggest relabeling the legend in Figure 5C to say LSH-KO instead of just KO.

R: We changed the label for improved clarity.

• Some signal seems to have been cut off in the edges of Figure 4E, cleaning up the image and/or showing the whole blot would be good.

R: We cleaned the image and the full blot will be available in the source file.

REVIEWERS' COMMENTS

Reviewer #1 (Remarks to the Author):

This is a vastly improved version of the manuscript. The authors have addressed all the suggestions and concerns. Specially adding RNAi experiments for the specific inhibition of different factors in addition to the chemical inhibitors initially studied here. The RNAi data provide additional support to the conclusions. The images of telomere defects are vastly improved and now provide sufficient detail to appreciate telomere defects.

Reviewer #2 (Remarks to the Author):

The authors have properly addressed all my concerns. The results now presented make the manuscript very solid.

Some typos need correction (e.g. Pag 18, BRAC1 spelling).

Reviewer #3 (Remarks to the Author):

I appreciate the thorough and complete response to reviews offered by the authors. I think the study is now much improved and I have no further critiques.

I will note that Figure 4D-E had some extra text overlaid (i.e. a small e under a big E; LSH overlapping with RAD51). Figure 7e had some bars where the stars indicating statistical significance seemed to be missing. This could all be a PDF conversion issue but the authors should check all their figures to ensure clarity in labelling.

We like to thank the reviewers for the helpful suggestions to improve the manuscript. Please see below for our point-by-point response to the reviewers' comments and concerns.

REVIEWER COMMENTS

Reviewer #1 (Remarks to the Author):

This is a vastly improved version of the manuscript. The authors have addressed all the suggestions and concerns. Specially adding RNAi experiments for the specific inhibition of different factors in addition to the chemical inhibitors initially studied here. The RNAi data provide additional support to the conclusions. The images of telomere defects are vastly improved and now provide sufficient detail to appreciate telomere defects.

Reviewer #2 (Remarks to the Author):

The authors have properly addressed all my concerns. The results now presented make the manuscript very solid.

Some typos need correction (e.g. Pag 18, BRAC1 spelling).

R: We corrected the typos.

Reviewer #3 (Remarks to the Author):

I appreciate the thorough and complete response to reviews offered by the authors. I think the study is now much improved and I have no further critiques.

I will note that Figure 4D-E had some extra text overlaid (i.e. a small e under a big E; LSH overlapping with RAD51). Figure 7e had some bars where the stars indicating statistical significance seemed to be missing. This could all be a PDF conversion issue but the authors should check all their figures to ensure clarity in labelling.

R: We checked the labelling and corrected the issues.